# Bubbles enable volumetric negative compressibility in metastable elastocapillary systems

Davide Caprini[1,9], Francesco Battista [2,9], Paweł Zajdel [3,9], Giovanni Di Muccio [2,9], Carlo Guardiani [2], Benjamin Trump[4], Marcus Carter[4], Andrey A. Yakovenko[5], Eder Amayuelas[6], Luis Bartolomé [6], Simone Meloni [7] ✉, Yaroslav Grosu [6,8] ✉, Carlo Massimo Casciola [2] ✉ & Alberto Giacomello [2] ✉

Although coveted in applications, few materials expand when subject to compression or contract under decompression, i.e., exhibit negative compressibility. A key step to achieve such counterintuitive behaviour is the destabilisations of (meta)stable equilibria of the constituents. Here, we propose a simple strategy to obtain negative compressibility exploiting capillary forces both to precompress the elastic material and to release such precompression by a threshold phenomenon – the reversible formation of a bubble in a hydrophobic flexible cavity. We demonstrate that the solid part of such metastable elastocapillary systems displays negative compressibility across different scales: hydrophobic microporous materials, proteins, and millimetre-sized laminae. This concept is applicable to fields such as porous materials, biomolecules, sensors and may be easily extended to create unexpected material susceptibilities.

Negative compressibility (NC) is a term used in literature to broadly indicate the unusual behaviour of materials expanding when compressed or contracting when decompressed. The term has been adopted for linear[1–4], area[1,5,6], or volume[7,8] NC. Such counterintuitive behaviour enables the realization of different applications in pressure sensors[1,9], acoustics[10], auxetic materials[11], artificial muscles[1], and pressure-modulated superconductors[12]. In this context, great promise resides in metamaterials[13,14] – materials that gain their properties from structure rather than composition[7] – which can be miniaturized by carefully designing crystalline nanoporous materials, such as Metal Organic Frameworks[15,16], which have indeed shown large NC[6,17,18].

A careful definition of NC is needed to distinguish cases in which the expansion upon compression in one direction is overcompensated by contraction in the other directions from those in which the volume of the system undergoes expansion[7], i.e., *the negative of the quantity defined as compressibility* – the definition used in this work. This is the most stringent case because, to ensure mechanical stability, thermodynamics forbids this kind of NC for closed systems in thermodynamic equilibrium (i.e., when the system is equilibrated for unlimited

[1]Center for Life Nano- & Neuro-Science, Istituto Italiano di Tecnologia, Viale Regina Elena 291, Rome, Italy. [2]Dipartimento di Ingegneria Meccanica e Aerospaziale, Sapienza Università di Roma, Via Eudossiana 18, Rome, Italy. [3]A. Chełkowski Institute of Physics, University of Silesia, ul 75 Pułku Piechoty 1, Chorzów, Poland. [4]Center for Neutron Research, National Institute of Standards and Technology, Gaithersburg, Maryland, USA. [5]X-Ray Science Division, Advanced Photon Source, Argonne National Laboratory, Argonne, Illinois, USA. [6]Centre for Cooperative Research on Alternative Energies (CIC energiGUNE), Basque Research and Technology Alliance (BRTA), Alava Technology Park, Albert Einstein 48, Vitoria-Gasteiz, Spain. [7]Dipartimento di Scienze Chimiche e Farmaceutiche, Università degli Studi di Ferrara, Via Luigi Borsari 46, Ferrara, Italy. [8]Institute of Chemistry, University of Silesia, Katowice, Poland. [9]These authors contributed equally: Davide Caprini, Francesco Battista, Paweł Zajdel, Giovanni Di Muccio. ✉e-mail: simone.meloni@unife.it; ygrosu@cicenergigune.com; carlomassimo.casciola@uniroma1.it; alberto.giacomello@uniroma1.it

A)
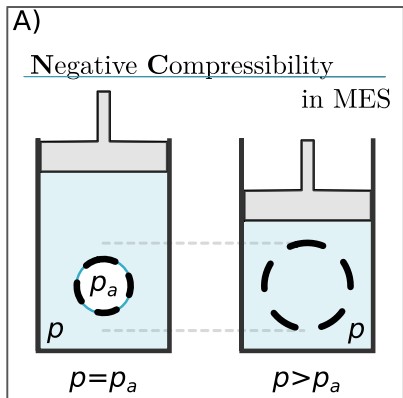

B)
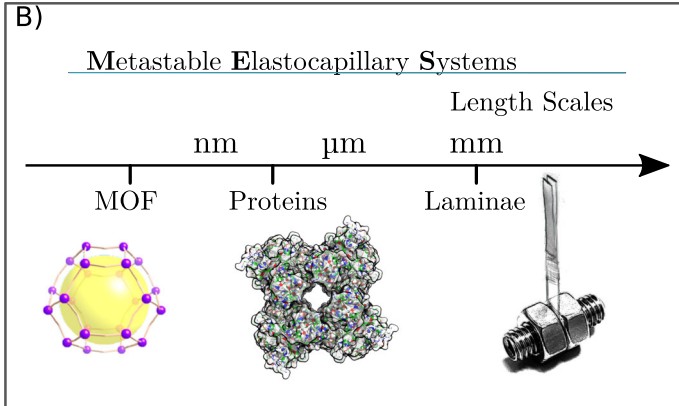

**Fig. 1 | Negative Compressibility. A** Sketch of the behaviour of Metastable Elastocapillary Systems (MESs) under hydrostatic pressure $p$, see also Supplementary Movie 1 showing the dynamic sketch of the negative compressibility. Negative compressibility is displayed by the solid part of MESs. $p_a$ is the ambient pressure; dashed lines represent the solid boundary and the thin blue line represents the gas-liquid interface; the light blue background represents water, while the gas filling the solid is in white. **B** Summary of the cross-scale and cross-domain concept of MES leading to systems with giant negative compressibility.

time)[19,20] as anticipated in the context of the Landau theory of phase transformations[21], see also[22]. In this work and in others[7], volumetric NC has been achieved by lifting the hypotheses on i) the kind of system or on ii) thermodynamic equilibrium by considering open systems or metastable states, respectively. Notable examples demonstrating the existence of volumetric NC are metamaterials exhibiting transitions between metastable solid phases[20] and the metastable elastocapillary systems (MESs)[8] considered here, whose solid component is an open system exhibiting NC.

Here, we introduce the concept of MESs and explain its working principle, showing that architectures ranging from subnanometric porous materials to millimetre-scale hydrophobic metamaterials and to nanometre-scale biological matter (ion channels) display a unified phenomenology (Fig. 1B). This mechanism is shown to originate in the dual role of capillarity: it compresses the elastic material before the liquid pressure is increased and it engenders first order transitions – here intrusion and extrusion of water in a hydrophobic cavity – that relax these deformations producing NC. Using these principles, we measured giant negative compressibility in the microporous material ZIF-67 (zeolitic imidazolate framework) and designed a millimetre-scale MES displaying a record NC exceeding $-10^7$ TPa$^{-1}$ and 18% relative deformation, which notably outperforms the previous record NC held by ZIF-8, $-1000$ TPa$^{-1}$ and 0.34% relative deformation[8]. The MES concept provides a simple, general, and cross-scale platform for engineering materials with negative compressibility and sheds additional light into the physics of protein transitions under pressure[23]. One should be careful to distinguish the MES NC transitions reported here that follow the rigorous definition given in Ref. 7 and illustrated in Fig. 1A, from other mechanisms, such as stretch-densification, that have been reported elsewhere[1,24]. In addition, at a variance with the continuous water uptake at the origin of sponge swelling, which alters the chemical interactions between cellulose nanofibrils or nanocrystals[25–27], the NC process discussed in this work is purely mechanical, corresponding to a first-order transition between a stable and a metastable state, i.e., confined liquid and confined vapour, while the chemistry of the solid medium is not altered. The reported NC mechanisms rely on reversible physical processes occurring at low pressures. In contrast, the pressure-induced formation of an expanded (super)hydrated phase in zeolites[25] was reported at high pressures (GPa); this transformation was retained after pressure release. Similarly, in sponge-like cellulose systems the swelling is associated to a more complex process, starting with a chemical interaction between water and cellulose hydroxyl groups, resulting into a partially irreversible structure modification of the system[26,27], that cannot be restored simply by releasing the pressure, but typically requires complete drying of the matrix. Finally, more complex behaviours were reported for other adsorbing materials when the mechanical pressure is combined with adsorption phenomena, e.g., the negative gas adsorption transitions in MOF[28–30]. As for the sponge-like systems, the adsorbed gas (the guest) modifies the molecular structure of the solid (the host), possibly expanding the solid while increasing the partial pressure of the gas. So, despite the apparent similarity with the MES phenomenology, one should note that the driving force of the expansion in the reported MOF systems is the chemical modification of the adsorbing matrix by the guest molecules. As for the sponge-like systems, such transitions are observed for hydrophilic porous materials and, hence, from an engineering point of view, their implementation in hydrostatic pressure applications is not straightforward.

## Results

In MESs comprising a flexible and hollow solid immersed in a non-wetting liquid, hydrophobic interactions support the existence of (meta)stable bubbles inside cavities, see Fig. 2A. However, sufficiently large liquid pressures may overcome capillary forces causing wetting of the cavities ("intrusion")[31]. In some cases, upon decompression, the bubble forms again ("extrusion"). A reversible intrusion-extrusion cycle, combined with the elasticity of the material, has been shown to give rise to MESs with giant NC[8,16]. We consider a paradigmatic MES consisting of two hydrophobic plates immersed in water and separated by a spring (Fig. 2A), for which the free-energy variation $\Delta\Omega$ is given by elastic, bulk (pressure), and surface contributions:

$$\Delta\Omega = \frac{1}{2}k(a - a_0)^2 + \phi_g(p - p_a)V + \alpha\gamma A, \qquad (1)$$

where $k$ is the spring constant, $a - a_0$ the variation of the interplate distance with respect to its resting value, $V$ the volume between the plates, $p$ and $p_a$ are the pressures of the liquid and the ambient pressure, respectively, $\gamma$ the liquid-gas surface tension, and $A$ the surface area enclosing $V$, see the Supplementary Text 1 for further details. The dimensionless parameters $\phi_g$ and $\alpha$ account for the volume and surface fractions occupied by and in contact with the liquid, respectively; both depend on $p - p_a$ that governs the progress of the intrusion and extrusion processes. Specifically, $\alpha$ recapitulates the budget of surface costs and gains related to the liquid-gas, liquid-solid, and solid-gas interfaces, expressed in terms of the dimensionless areas $\phi_{lg} = A_{lg}/A$ and $\phi_{sg} = A_{sg}/A$ occupied by the liquid-gas and solid-gas interfaces, respectively: $\alpha = \phi_{lg} + \phi_{sg}\cos\theta_Y$, with $\cos\theta_Y \equiv (\gamma_{sg} - \gamma_{sl})/\gamma$

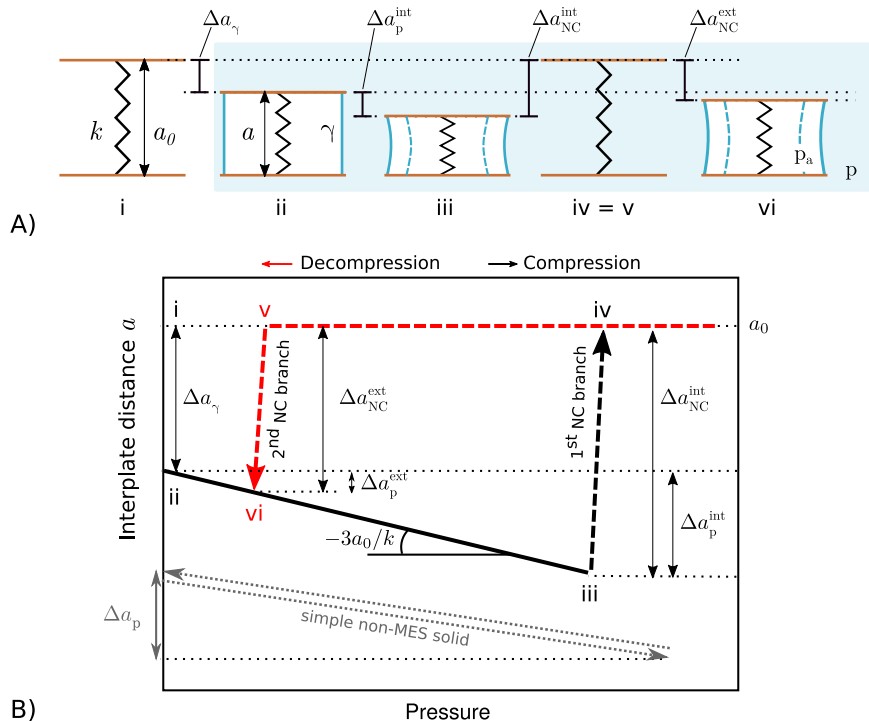

**Fig. 2 | Model of a metastable elastocapillary system exhibiting negative compressibility. A** Sketch of a thought intrusion and extrusion experiment in which (i) two plates (brown) kept together by a spring (black) of constant $k$ and length $a$ (equilibrium length $a_O$) are immersed in water (ii), then the hydrostatic pressure $p$ is progressively increased (iii) until intrusion occurs (iv). The pressure is subsequently decreased from the fully intruded state, until extrusion is achieved (vi) and the cycle can repeat. $p_a$ is the ambient pressure; $\Delta a_{NC}^{int}$ and $\Delta a_{NC}^{ext}$ are the negative compressibility jumps at intrusion and extrusion, respectively, and $\Delta a_p$ and $\Delta a_\gamma$ denote the elastic compression and the capillary precompression contributions to such jumps, respectively, see main text. Blue lines represent the gas-liquid interface, with $\gamma$ being the surface tension. **B** Representation of the cycle in the pressure–interplate distance plane. Grey dashed lines indicate the compression/decompression cycle for a simple solid material (non-MES).

the Young contact angle. Upon intrusion, liquid penetrates into the hydrophobic voids leading to $\phi_g = \alpha = 0$; the pressure $p_{int}$ at which this intrusion process happens can be included in the model above by providing a detailed wetting mechanism[32] or, as done here, specified externally. Similarly, the extrusion process is included here on an ad hoc basis. The system described by Eq. (1) is bistable with a trivial solution $a = a_0$, corresponding to the fully wet cavity, and a solution which depends on the applied pressure, corresponding to the dry cavity immersed in water. Intrusion and extrusion phenomena allow the system to switch between these states, conferring reversible NC to the system.

When a cycle is performed by increasing the hydrostatic pressure and subsequently decreasing it, the model in Eq. (1) yields the following behaviour, see Fig. 2: i) (before the actual experiment starts) the hydrophobic cavity is in air in the resting state $a = a_0$; ii) capillary "precompression" occurs upon contact with water, at $p = p_a$; ii-iii) pressure further contracts the cavity (positive compressibility branch) until iii-iv) liquid intrudes into the cavity, suppressing the liquid-gas interfaces, setting $\phi_g = 0$, and relaxing elastic deformations (first NC branch); iv-v) the solid is in the resting state, without deformations upon hydrostatic compression/decompression, because liquid is present both inside and outside the cavity, exerting the same pressure (horizontal branch); v-vi) liquid extrudes from the cavity (second NC branch), which is possible only when the intrusion phenomenon is somewhat reversible, e.g., in the case of vapour nucleation[31,33]. Nanoporous MESs are indeed fully reversible because the extreme hydrophobic confinement can induce drying[32]; depending on the specific system, extrusion may occur at positive or negative $p$[34]. We remark that, even if the solid part of MESs presents substantial NC, the overall system, which includes the liquid, does not exhibit such feature, see the position of the piston in Fig. 1A.

Our simple model offers a general explanation of NC in MESs. Any standard elastic solid undergoes contraction or expansion upon hydrostatic compression or decompression, respectively, without NC, as shown by thin grey lines in Fig. 2B; this elastic deformation reads $\Delta a_p/a_0 = -3a_0(p - p_a)/k$ (for details see the Supplementary Text 1). In the sudden expansion of a MES at $p_{int}$, which defines NC, this trivial contribution $\Delta a_p$ is present but the crucial contribution $\Delta a_\gamma$ is of capillary origin: $|\Delta a_{NC}^{int}| = \Delta a_p^{int} + \Delta a_\gamma$. Indeed, Fig. 2B shows that $\Delta a_\gamma$ proceeds from the release of the capillary precompression due to formation of liquid-gas interfaces when the hollow hydrophobic solid is placed in contact with the liquid at $p = p_a$ (cf. points i and ii in Fig. 2A). The contribution $\Delta a_\gamma/a_0 = -2\alpha\gamma/k$ is clearly present only in MES and accounts for their NC (see also the Supplementary Text 1). Analogously, in the extrusion branch, nucleation of liquid-gas interfaces induces a contraction $|\Delta a_{NC}^{ext}| = \Delta a_p^{ext} + \Delta a_\gamma$; the non-trivial capillary contribution is more prominent compared to intrusion because the extrusion pressure is lower than the intrusion one, Fig. 2B.

Remarkably, the MES features reported for our simple model and highlighted in Fig. 2 were found in the diverse systems we analysed, i.e., nanoporous ZIF-67 materials (Fig. 3), hydrophobic laminae (Fig. 4), and biological ion channels accounting for a unified explanation of NC in elastocapillary systems undergoing pressure changes. The important performance and design parameters characterising these MESs, including NC, are identified in our model and reported in Table 1 for the three cases.

The first MES we tested is made of a microporous, hydrophobic, and flexible crystalline material immersed in water, ZIF-67[35] (sub-nanoMES, Fig. 3). MES based on nanoporous materials are particularly promising because they exhibit extremely large surface areas per unit mass, ca. 1600 m²/g for ZIF-67 (Supplementary Fig. 2), are easy to miniaturise (nanoZIF-8[36]), and their properties can be tailored by

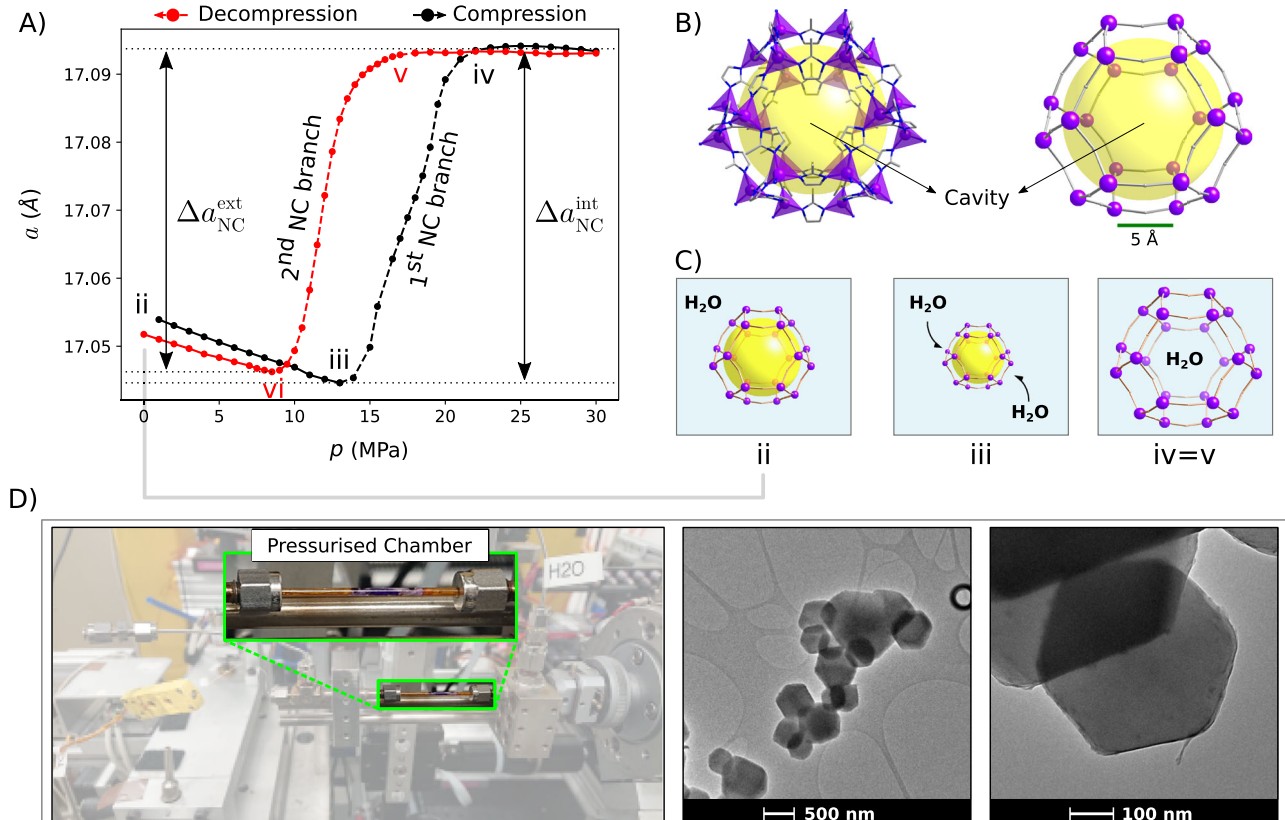

**Fig. 3 | Intrusion/extrusion experiment in ZIF-67. A** Synchrotron data allow to track the changes in the lattice parameter $a$ as the hydrostatic pressure in the water/ZIF-67 system is increased from low values to above the intrusion pressure (black) and back to ambient values, triggering extrusion (red). The system display the two NC (negative compressibility) transitions, in agreement with the model shown in Fig. 2. Data were collected at beamline 17-BM at the Advanced Photon Source, Argonne National Laboratory. Source data are provided as a Source Data file. **B** Molecular structure (left) and topological simplification scheme (right) of ZIF-67 (Zeolitic imidazolate framework); violet denotes cobalt ions and the related tetrahedra connected by **imidazolate** linkers; the yellow sphere highlights the cavity.

**C** Conceptual scheme of the behaviour of a single cage of ZIF-67 during the initial compression (ii-iii) and following water intrusion (iv=v). Blue background stands for water; yellow highlights the empty cavity. **D** Experimental setup and TEM image of the ZIF-67 nanoparticles. The sample is loaded into a single crystal sapphire capillary with a K type thermocouple inside the sample. The sample is composed of nanoparticles having an average size of $509 \pm 13$ nm (standard deviation 124 nm). TEM measurements were performed on a Tecnai G2 F20 Super Twin under 200 kV acceleration voltages; sample powder was dispersed in **ethanol**, sonicated using a water bath and placed in a holey carbon grid.

acting on the porous material[37], on its connectivity[34], by carefully designing its crystalline (Supplementary Fig. 1) structure[38], or by changing the intruding liquid. ZIF-67 consists of cages separated by 8 subnanometric windows (Fig. 3B), which play a crucial role in the intrusion process[8]. We measured deformations of ZIF-67 during the $H_2O$ intrusion/extrusion cycle via in operando synchrotron X-ray diffraction adopting a specially designed wet cell with adjustable pressure (Fig. 3D). Data revealed pronounced NC: a sharp increase of the lattice parameter upon compression due to intrusion and its decrease upon decompression due to extrusion. Due to its cubic symmetry, the measured lattice parameter increment is directly translated into the volumetric NC of ZIF-67 (Fig. 3A, C, from state iii to iv). This finding is repeatable and consistent with the $D_2O$ intrusion-extrusion cycle for ZIF-8 previously explored in terms of VNC[8]. Notwithstanding the subnanometric pore openings, the intrusion pressure is relatively low, ca. 23 MPa, due to the flexibility of the material, which facilitates the opening of intercage windows[36]. We remark that the pressures used in our experiments, $p < 30$ MPa, are much lower than the pressures needed to induce structural phase transitions in ZIF-8[39,40], which are of the order of one GPa. Also extrusion is expected to be facilitated by flexible materials[41], which indeed occurs at ca. 9 MPa for ZIF-67, ensuring reversibility of the cycle.

Figure 3 shows that the subnanoMES has all features described by our model, including a substantial capillary precompression $\Delta a_\gamma/\Delta a_{NC} \approx 0.8$ and 0.9 upon intrusion and extrusion, respectively. This term is indeed expected to dominate $\Delta a_{NC}$ in nanoporous MESs due to the favorable scaling of capillary terms as compared to volume ones at the nanoscale $\Delta a_\gamma/\Delta a_p \propto \gamma/(a_0(p-p_a)_{int/ext})$. Because of hydrophobic aggregation of the porous grains upon mixing with water at ambient pressure an independent measure of $\Delta a_\gamma$ was never achieved but can be deduced by comparing Figs. 2 and 3. The measured NC is giant for ZIF-67, $\beta_{1D}^{int} = 613$ TPa$^{-1}$ and $\beta_{1D}^{ext} = 815$ TPa$^{-1}$, due to the rather abrupt intrusion and extrusion processes, see Table 1. On the other hand, the maximum changes in the lattice parameter in intrusion and extrusion are ca. 0.29% of the resting value. Overall, the nanoporous MES displays giant NC both in intrusion and extrusion, both occurring over a narrow pressure range; in addition, the pressure cycles are completely reversible with limited pressure hysteresis. The work that can be performed upon compression can be roughly estimated by multiplying $p\Delta V$ during the intrusion (1$^{st}$ NC branch) for a single unit cell and taking into account the average unit cells per gram, which yields 0.04 J/g.

Not only does the model in Eq. (1) fully clarify the behaviour of the current and previous[8,16] subnanoMESs, including capillary precompression, but it has far more general implications for NC. Importantly, Eq. (1) applies to MES of any realm and size. The crucial element at the heart of the reported behaviour is capillary precompression which gives rise to NC−the unusual expansion of the solid part of MESs

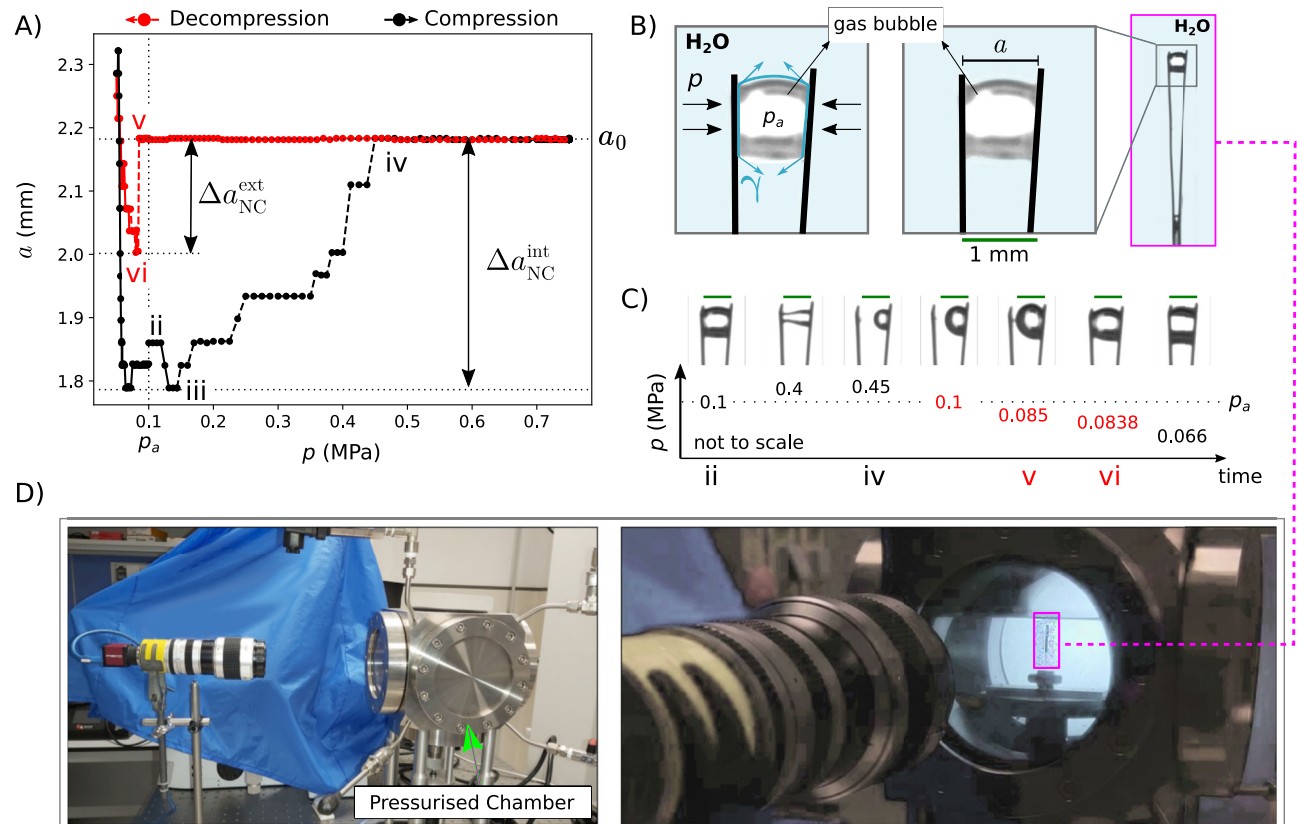

**Fig. 4 | Intrusion and extrusion experiment in the milliMES (millimitre metastable elastocapillary systems) composed of a pair of facing hydrophobic laminae immersed in water. A** Variation of the distance $a$ between the tips of the laminae vs the liquid pressure. Black denotes compression and red decompression. The area between the curves represents the system work and is estimated $3 \cdot 10^{-3}$ J per laminae ~ 0.15 J/g, considering the mass of a single Teflon laminae, 0.02 g. Source data are provided as a Source Data file. **B** Zoom on the milliMES tip displaying the presence of a vapour bubble, with details of the system: left panel shows the pressure forces (black arrows) and surface tension (blue) acting on the laminae;

in the central panel, the distance $a$ and the scale bar (1 mm, green) are indicated; the left panel shows a snapshot of the laminae in its full length. **C** Snapshots of the system for selected pressures showing the different configurations of the milliMES. The green scale bars are 1 mm. The single foil is 15 x 2 x 0.2 mm$^3$. See Supplementary Movie 2 for the full recorded video. **D** Two views of the experimental setup. The hydrophobic laminae are contained in a pressurised chamber whose pressure can be controlled externally. Deformations during the compression/decompression experiment are recorded by an external camera.

**Table 1 | Parameters characterising the different MESs (metastable elastocapillary systems) considered in this work**

| System | $p_{int}$ | $p_{ext}$ | $a_0$ | $\frac{\Delta a_{NC}^{int}}{a_0}$ | $\frac{\Delta a_{NC}^{ext}}{a_0}$ | $\frac{\Delta a_p^{int}}{a_0}$ | $\frac{\Delta a_p^{ext}}{a_0}$ | $\frac{\Delta a_\gamma}{a_0}$ | $\beta_{1D}^{int}$ | $\beta_{1D}^{ext}$ |
|---|---|---|---|---|---|---|---|---|---|---|
| ZIF-67 | 23 | 8.72 | 17.053 | 0.29 | $-0.28$ | $-0.059$ | 0.032 | $-0.23$ | $-286$ | $-269$ |
| ZIF-8 | 31 | 20.3 | 17.09 | 0.34 | $-0.34$ | $-0.090$ | 0.074 | $-0.25$ | $-841$ | $-487$ |
| Laminae | 0.45 | 0.985 | $2.18 \cdot 10^7$ | 17.9 | $-8.3$ | $-3.2$ | - | $-14.7$ | $-5.8 \cdot 10^5$ | $-4.1 \cdot 10^7$ |
| MscL | >40 | - | 25.23 | >40 | - | $-0.2$ | - | - | $-1 \cdot 10^4$ | - |
| BK | 100 | - | 39.4 | 16.8 | - | $-0.6$ | - | - | $-1680$ | - |

The reported 1D compressibilities are defined as $\beta_{1D} = -\Delta a/a_0/\Delta p$, where $\Delta p$ refers to the pressure drop needed to achieve the displacement $\Delta a$; this monodimensional definition is chosen to facilitate comparison of different systems and underestimates NC as compared to the conventional one based on the maximum derivative da/dp, used, e.g., in Ref. 8. ZIF-67 values at 5˚ C. ZIF-8 data from[8], with $\beta$ recomputed according to the present definition. Biological pores data (MscL and BK channels) are inferred from different works and are reported for comparison, see Supplementary Text 2 for details. Missing data are denoted by a dash. Table units: $p_{int/ext}$ (MPa), $a_0$(Å), $\Delta a/a_0$ (%), $\beta_{1D}^{int/ext}$ (TPa$^{-1}$).

upon intrusion and their contraction upon extrusion. Metastabilities in the system allow the reversible switching between the confined liquid and gaseous states triggered by pressure. Using these simple guidelines, in the following, we devise a simple millimetre-scale MES (milliMES), i.e., 6 orders of magnitude larger than ZIF-67, and we attempt to recognize possible MESs in biological pores having nanometre size.

We realised millimetre-sized hydrophobic metamaterials (Fig. 4), consisting of pairs of facing laminae, cut out from a Teflon film, which makes them intrinsically hydrophobic. Owing to their hydrophobicity and the small intervening space, an air bridge is formed between the laminae upon immersion in water, similar to the empty ZIF-67 cages.

Our idea is to elastically deform the laminae by capillarity and pressure and restore the resting state when the air bridge is suppressed at $p_{int}$; this mechanism is expected to produce NC. First, in an air-tight cell (Fig. 4D), we increased the water pressure from ambient value until ca. 0.7 MPa. Subsequently, we opened a valve to decrease the pressure to the ambient value and then connected the cell to a vacuum pump to reach values slightly below $p_a$. Results reported in Fig. 4A show a remarkable similarity to the systems in Figs. 2 and 3: an extended NC branch upon intrusion, a horizontal branch corresponding to the wet laminae, and an unexpected NC branch at $p < p_a$, which corresponds to the recovery of the initial air bridge.

The snapshots of the system at different pressures in Fig. 4C reveal in detail the operating mechanism of the milliMES. Positive compressibility of the system is hardly noticeable when the pressure is first increased because of the partial compensation of the two forces acting on the laminae: pressure and capillary forces (Fig. 4B), which act at the contact line of the air bridge with the laminae; when pressure is increased, the first contribution increases but the triple lines at the lower end moves upwards, causing a progressive decrease of capillary forces. Because of this competition, the NC branch has a mild slope, except in the final part which is characterised by a sudden jump corresponding to the suppression of the air bridge at $p = 0.4$ MPa. This process is equivalent to liquid intrusion in ZIF-67. When water fills the gap between the two laminae, they become disassociated and their distance does not vary upon increasing or decreasing the pressure, similarly to Figs. 2 and 3; however, two asymmetric bubbles, remnants of the original air bridge, are still present on individual laminae. At pressures below the ambient one, these air bubbles grow substantially, eventually merging again in an air bridge at $p = 0.985$ MPa. Although the microscopic mechanism is rather different from the vapour nucleation giving rise to extrusion in the subnanoMES, the net effect is similar: it produces a sharp NC jump under tension and it allows to form again the liquid-gas interfaces which are needed to precompress the MES and restart the cycle (Supplementary Fig. 4).

The air bridge grows upon further decreasing the pressure (growing branch with negative slope in Fig. 4A) and rapidly shrinks (decreasing branch with negative slope) when the pressure is increased, closing the cycle, which is largely reversible and repeatable (Supplementary Fig. 4). We highlight that the intrusion and extrusion cycles present some variability among independent replicas (Supplementary Fig. 4B), originating in i) the pinning phenomena due to fine details of the surface and geometry of the laminae and ii) the initial conditions of the bubble. This variability could be mitigated by using more careful designs of hydrophobic laminae or by automated manufacturing and experimental procedures. The resulting maximum relative deformation of this MES is ca. 18% and $\beta_{ID}^{int} = -5.8 \cdot 10^5$ and $\beta_{ID}^{ext} = -4.1 \cdot 10^7$ TPa$^{-1}$. These NC values are surprisingly up to 4 orders of magnitude larger than the current record NC held by subnanoMES[8], reflecting the fact that using softer materials can help maximising NC; furthermore, $\beta_{ID}^{ext}$ is sensibly larger because of the abrupt phenomenon of bubble merging. The milliMES exhibits the generic features predicted by our model and already verified for the subnanoMES: reversible compression/decompression cycles with two NC branches. Quantitative differences from the subnanoMES are the larger relative deformations and the more gradual NC branch for intrusion. The work that can be performed upon compression can be estimated for a single pair of laminae to be 0.002 J.

Having demonstrated the MES concept for elastocapillary systems differing in size by 6 orders of magnitude, we ask ourselves whether the same paradigm can hold also for more complex, biological systems. Protein expansion under high hydrostatic pressure is a known phenomenon[23], which has been proposed to be related to the intrusion of water inside the hydrophobic core[42]. Here, we specifically target ion channels that have important analogies with the subnanoMES. These pore-forming proteins respond to external stimuli to regulate the flux of ions across cellular membranes[43]. Some of them exhibit hydrophobic gating, i.e., the stochastic formation/destruction of a "bubble" in (sub)nanometre-sized hydrophobic sections of their pores; this event switches off/on the passage of ions which cannot translocate through the ion channel without their hydration shell[44,45]. Notwithstanding the increased structural and chemical complexity of ion channels and of their environment, hydrophobic gating bears important analogies to intrusion/extrusion in hydrophobic nanopores[31], suggesting that pressure should favor the destruction of the nanoscale bubble and may trigger the opening of the channel.

Channel opening is typically accompanied by an expansion of the ion channels, which would thus result in NC.

Very few experiments are available on the functioning of ion channels at high hydrostatic pressures, see Supplementary Text 2; among these, the bacterial mechanosensitive channel MscL[46] (Supplementary Fig. 5) and the BK potassium channel[47] (Supplementary Fig. 6) exhibit reversible channel opening in response to high hydrostatic pressures (ca. 90 MPa). This behaviour was reportedly at odds with Le Chatelier principle[46], according to which ion channels should contract upon compression (i.e., positive compressibility). However, based on the present results and on the fact that hydrophobic gating has been reported for both MscL[48,49] and BK[50], we propose that both ion channel may behave as biological MESs with NC. The values inferred from patch clamp experiments under high hydrostatic pressure are reported in Table 1 for comparison with the other MESs, highlighting that, because of its flexibility, biological matter can achieve extreme NC even at the nanoscale. It should be noted that ion channels are in several ways more complex than our simple model; in particular, they interact with the lipid bilayer, respond to voltage and ligands, etc.

## Discussion

Figure 1 summarises the MES concept and results: systems spanning more than 6 orders of length scales and different realms share a single unifying principle for NC: capillary precompression in hydrophobic systems. In principle, this range could be further extended in microgravity conditions by removing the upper limit on the considered capillary phenomena, set by the capillary length $l_c = \sqrt{\gamma/(\Delta\rho g)} \approx 2$ mm, with $\Delta\rho$ the mass density difference between water and air and $g$ the gravitational acceleration. In MESs, metastabilities provide a reversible threshold phenomenon (intrusion/extrusion) which releases elastic deformations of capillary origin upon compression and decompression, producing NC. Simple elastic materials without the combination of metastabilities and capillarity cannot show this mechanism. Reversibility was surprisingly observed in all systems[51], although its origin is different: confinement-assisted nucleation of water vapour at the subnanoscale[31] and merging of air bubbles at the millimetre scale. Strikingly, this working principle of MESs seems common to man-made systems and to biological ones; in particular, both elastic nanoporous materials and ion channels seem to feature similar NC mediated by capillary forces.

Table 1 compares the different MESs considered in this work and from the literature, demonstrating their considerable design flexibility. Firstly, MESs can be realized at different scales. Secondly, the intrusion and extrusion processes, which determine the value of NC, can occur in a more or less abrupt fashion adapting to different applications: a fast NC response is important for switches whereas a more progressive NC for compensating positive compressibility of a device. In the considered MESs, NC is rather sharp for ZIF-67, in which vapour bubbles easily absorb and nucleate, while wetting is more gradual for hydrophobic laminae where the air bridge slowly changes morphology; for nanoporous materials, this aspect could be further optimized, e.g., by mixing different kinds of porous material with different $p_{int/ext}$. Thirdly, the MES platform is based on rather general mechanical principles which could be used to create unexpected material susceptibilities, e.g., inducing intrusion and extrusion by different driving forces, such as temperature, electrowetting, solvent concentration, etc., to provide unexpected negative coefficients as the previously reported negative thermal expansion of microporous materials[52]. For specific applications, the MES properties and their variability could be tailored by carefully controlling sample quality[53] (subnanoMES) or automated manufacturing and experimental procedures (milliMES).

In conclusion, metastable elastocapillary systems provide a unified, cross-scale platform to develop materials with negative

compressibility and to understand their occurrence in nature. Based on this understanding, we designed and tested a milliMES, which sets the current record for negative compressibility while exhibiting very large relative changes in dimension. On the application side, MESs provide a way to combine in the same system a pressure sensor and a threshold switch. This mechanism is easy to generalize to sensors/switches of temperature, concentration, and humidity. In biology, the MES paradigm may help understanding important aspects of molecular adaptation of deep-sea organisms while suggesting additional ways to trigger controlled cell response, e.g., by high-intensity focused ultrasound.

## Methods

**Disclaimer** Certain commercial equipment, instruments, or materials (or suppliers, or software, …) are identified in this paper to foster understanding. Such identification does not imply recommendation or endorsement by the National Institute of Standards and Technology, nor does it imply that the materials or equipment identified are necessarily the best available for the purpose.

### ZIF-67

**Materials.** All solvents and chemicals were used as received from reliable commercial sources. **Cobalt (II) nitrate hexahydrate** ($Co(NO_3)_2$) and **methanol** 96% were purchased from Fisher Scientific, **2-methylimidazole** (Hmim) was purchased from Sigma-Aldrich Co. and **ethanol** (technical grade) was purchased from Scharlab S.L.

**ZIF-67 synthesis.** Synthesis of ZIF-67 was based on the protocol reported by H. Wu et al.[35] with slight adaptations. Two separate solutions of $Co(NO_3)_2 \cdot 6H_2O$ 80-mM and Hmim 320 mM were prepared in 100 mL of **methanol**. Both solutions were then mixed and stirred for one minute and the resulting mixture was left crystallising at room temperature for 48 h. The supernatant was removed, and the purple precipitates were collected by centrifugation and washed thoroughly with ethanol three times. The final product was dried at 80 °C overnight.

**X-ray diffraction.** Structural characterisation of the ZIF-67 samples was performed by means of X-ray diffraction at room temperature in powdered samples using a BRUKER-D8 ADVANCE X-ray diffractometer using CuK$\alpha$ (0.15406 nm) recorded in $2\theta$ steps of 0.02° in the $5 - 80°$ range.

Experimental XRD pattern compared with the calculated one shows a good matching of all ZIF-67 reflections, confirming structural viability and purity of the synthesised sample (Supplementary Fig. 1).

**Nitrogen adsorption.** Textural characterisation was carried out in an automated gas adsorption analyser (Micromeritics ASAP 2460). **Nitrogen** physisorption curves were recorded at 77 K (Supplementary Fig. 2, after outgassing at 200°C in vacuum for 15 h. The specific surface area (SBET) was calculated using the BET equation in the linearised version proposed by Parra et al.[54]. Total pore volume was calculated as the volume of gas adsorbed at a relative pressure of 0.99 in terms of $cm^3/g$, using the density of liquid gas to convert the amount of gas adsorbed in STP conditions with the expression $V_{(TOTAL\ PORES)}$ ($cm^3\ g^{-1}$)= $V_{(gas\ adsorbed)}$ ($cm^3\ g^{-1}$,STP) × 0.00154643.

**In operando synchrotron powder diffraction.** In operando pressure synchrotron powder diffraction studies were carried out at beamline 17BM of the Advanced Photon Source (APS, Argonne, USA). Si(311) monochromator was tuned to an incident energy of 27 keV. The resulting wavelength ($\lambda = 0.451910(1)$Å) and the detector distance (600-mm) were calibrated using NIST SRM660a LaB$_6$. Diffraction images were collected using Varex 3434CT 2D detector and processed

by GSAS-II[55] package. Powdered sample of ZIF-67 was loaded into a thin wall Al capillary with a K-type thermocouple inserted into the powder but outside the region of the beam. The angular regions with contributions from reflections from the capillary at $2\theta = 12.60°$, 14.40° and 18.10° were excluded from analysis. The sample was held in place with pieces of kapton tube and quartz wool. The temperature was maintained using Oxford Cryosystems Cryostream nitrogen blower on the basis of the readout from the thermocouple. Pressure in the system was dynamically stabilised by an ISCO syringe pump which was filled with Water ASTM Type II (VWR Chemicals BDH, BDH1168-4LP). The sample was activated in situ by flowing He gas at 90 °C for 15min. Then the sample was cooled down to ambient temperature and flooded with water.

Several XRDs collected before the activation, during activation, post activation at RT and after flooding with water at the RT did not reveal any signs of decomposition. The measurements were carried out by collecting 4 frames (0, 1, 2, 3) at each pressure. Each frame lasted 30sec and the first 3 frames (0, 1, 2) were collected when system was equilibrating to isothermal conditions. Only the frames number 3 were used for the analysis. The data were collected with 1 MPa step before and after the intrusion, while 0.5 MPa step was used in the intrusion/extrusion region. The effective pressurisation and depressurisation rates were 0.39 MPa/min pre- and post-intrusion and 0.19 MPa/min during the (in/ex)trusion. Pattern matching analysis was carried out using FullProf suite of programmes[56] (May 2021) and the serial refinements were automated using Python scripts.

The LeBail analysis was used to extract lattice parameters and estimate effects of peak broadening due to domain size effects and strain induced by the (in/ex)trusion. The coherent domain size and strain were estimated using models built-in into the Fullprof suite. The instrumental resolution file was obtained from the LaB6 standard used for calibration. The diffraction line profile was parameterized using the Thompson-Cox-Hastings parametrisation of the pseudo-Voigt profile. Although the original physical background of the TCH profile is not fully reflected in the 2D detectors[57], it still provides very good parameterisation of the whole diffraction profile. The size and size broadening is imposed on top of the machine resolution, so in the first-order approximation it does not depend on a particular baseline shape. The anisotropic strain broadening was parametrised in the quartic form and the anisotropic Lorentzian size broadening was modelled using spherical harmonic approach. The models appropriate for the Laue class *m-3m* (Size=17, Strain=13) were used and the effective size and strain were calculated by the FullProf.

### Hydrophobic laminae

**Preparation of the system.** Pairs of hydrophobic laminae are obtained by cutting the two foils from a Teflon sheet of a 200 $\mu$m thickness. The size of a single lamina is 20 mm long and 2 mm wide. The cutting of the foils is manually executed with a surgical scalpel to avoid edge imperfections. The perfectly rectangular shape of the sheets is ensured by means of a plastic reference template. The sheets are glued under a microscope with **cyanoacrylate** on a plastic **acetate** film of calibrated thickness to separate them by 500 $\mu$m in the present experiments. The thickness of the plastic strip determines the distance between the laminae at the root, whilst it is about 1 mm at the tip due to the residual divergence of the Teflon sheets. The final length of the laminae is 15 mm since the rest of their length is used for the bonding. This system is maintained in the desired position, in line with the porthole and the optical axis of the camera, by a support made of two metal hexagons with nut screws laid down on a metal plate.

**Experimental setup.** The setup is made of a steel chamber obtained from a cubic block and equipped with circular quartz portholes to allow optical access and several access doors to allow connections with

external pumps. The chamber is filled with degassed water, while sensors allow monitoring of the internal pressure. Experimental images are acquired by a monochrome camera (Allied Vision Mako U-130B) that, through a $18 - 108/2.5$ zoom lens, focuses in the chamber where the hydrophobic foils are immersed. The resolution of the camera is $1280 \times 1024$. To increase the contrast of the image a background light (LED matrix with diffuser) is employed on the opposite side of the camera, see the sketch in Supplementary Fig. 3A.

**Experimental procedure.** The hydrophobic laminae are immersed in water, assuring that air bubbles remain trapped between laminae from the free surface. To purify the water by dissolved gas, a boiling process is applied by using a microwave oven for about 20 min. The entire volume of about 3 l of water (and a small amount of air) and the samples are sealed inside the high-pressure test chamber. The chamber pressurization is obtained by pumping compressed air on the small free surface of the water in the upper cap of the chamber. The pressure level, the pressure increase rate, and its asymptotic value are controlled by a regulator which takes from the main compressed air line at 8 bar. To enforce the decompression up to the atmospheric pressure, a metering valve is present. To decrease the pressure below the atmospheric value, the chamber is also connected to a vacuum pump through a three-way valve. After reaching the minimal pressure, the metering valve allows the successive increase of the pressure up to the atmospheric value. The pressure signal is acquired at a 1 Hz frequency with a manometer ranging from $-1$ bar to 10 bar. In Supplementary Fig. 3B, a scheme of the experimental setup is reported.

### Reporting summary

Further information on research design is available in the Nature Portfolio Reporting Summary linked to this article.

## Data availability

All data needed to evaluate the conclusions in the paper are present in the paper and/or in the Supplementary Information. All the data shown are available with this paper and in the Zenodo database 11047018. In particular, the source data related to the following panels - ZIF-67: Fig. 3A; Supplementary Fig. 1; Supplementary Fig. 2. - Laminae: Fig. 4A; Supplementary Fig. S4A; Supplementary Fig. S4B. are provided as individual files, for a total of 6 files. The headers of the columns in each file report the name of the $x$ and $y$ axes of the respective graphs, and all the necessary information to interpret and analyse the data. Source data are provided with this paper.

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

## Acknowledgements

This project has received funding from the European Research Council (ERC) under the European Union's Horizon 2020 research and innovation program (grant agreement No. 803213) (AG). The project leading to this application has received funding from the European Union's Horizon 2020 research and innovation programme under grant agreement No. 101017858 (YG, SM). This project has received funding from the European Union NextGenerationEU/PRTR and MICIN/AEI/10.13039/501100011033, grant RYC2021-032445-I (YG). AG acknowledges financial support by the Italian Ministry of Education, University, and Research (MIUR) through the "Framework per l'Attrazione e il Rafforzamento delle Eccellenze per la Ricerca in Italia (FARE)" scheme, grant SERENA n. R18XYKRW7J. This research received financial support based on Decision No. 2021/43/D/ST5/00062 from the National Science Center, Poland (YG). This research used resources of the Advanced Photon Source, a U.S. Department of Energy (DOE), Office of Science, Office of Basic Energy Sciences, User Facility operated for the DOE Office of Science by Argonne National Laboratory under Contract No. DE-AC02-06CH11357 (PZ, BT, MC, AY). Extraordinary facility operations were supported in part by the DOE Office of Science through the National Virtual Biotechnology Laboratory, a consortium of DOE national laboratories focused on the response to COVID-19, with funding provided by the Coronavirus CARES Act (PZ, BT, MC, AY).

## Author contributions

Conceptualisation: A.G., Y.G., S.M., and C.M.C. Investigation: D.C., F.B., A.G., P.Z., G.D.M., B.T., M.C., A.Y., E.A., and L.B. Methodology: D.C., P.Z., A.G., C.G., Y.G., and C.M.C. Visualisation: D.C., G.D.M., F.B., and E.A. Supervision: A.G., C.M.C., Y.G., and S.M., Writing–original draft: A.G., F.B., and G.D.M., Writing–review & editing: all the authors.

## Competing interests

The authors declare no competing interests.
