## [Peer Review File · Nature Communications]

Bubbles enable volumetric negative compressibility in metastable elastocapillary systemsREVIEWER COMMENTS

Reviewer #1 (Remarks to the Author):

Manuscript Metastable elastocapillary systems with negative compressibility by Davide Caprini et al.

Authors describe a macroscopic laminae system of Teflon foils interacting with water under varied pressure conditions. There are several unclear points in this paper. The technical side of the experiment is not clear to me, and particularly the system change at point VI (page ??), between stages e-f in Figure 2. I would expect that when pressure is released water would stay between the films, unless the pressure reduced to vacuum – so all water evaporates. This makes the system operation really complicated. However, my main objection is the false use of term compressibility, which is very well defined in thermodynamics, as the negative volume change dV divided by (V multiplied by the pressure change dp). This definition must be applied for system under pressure, usually a uniform material. The reverse of compressibility is bulk modulus, characteristic of different materials. In the manuscript, Authors consider the volume of a sub-system in the bigger system under varied pressure. It should be made clear that the large-system volume is positively compressed, while the negative compressibility is claimed for the sub-system. It is not true, because its (sub-system) elements are not compressed at all – the films are pressurized from both inner and outer sides. Manuscript's SI video 2 showing a ball compressed in a liquid illustrates this misconception: in the left film the normal (positive compressibility) situation is shown, but the right film suggests that the negative compression is possible. But this right video neglects the intrusion of liquid (water) into the yellow ball. So the ball material transforms into another material.

These serious mistakes invalidate this paper. There is no negative compressibility at all. For this reason I cannot recommend this paper for publication.

Reviewer #2 (Remarks to the Author):

The paper “Metastable elastocapillary systems with negative compressibility” offers exciting new results on how to generate negative compressibility in metastable systems. Negative compressibility, as precisely defined in this manuscript, cannot occur continuously in thermodynamical equilibrium but has been previously shown to occur through phase transitions. In that case, the system transitions from one stable/metastable state to another in response to increased stress.

Here the authors explore another mechanism that can lead to negative compressibility, which is nicely illustrated in Fig 2a. The mechanism involves intrusion and extrusion of fluid particles under hydrodynamical compression/decompression. As pressure is increased, intrusion causes the material to expand.

The paper is very well written, with a good balance of experiments and numerical + analytical description. The paper is technically correct, and the results support the conclusions. The figures are clear, and the literature is suitably covered. Supplemental materials are relevant and should be published with the paper.

I recommend publication in Nature Communications, provided the authors can address the following minor point in their revisions:

In p. 2, the authors appear to distinguish the mechanism in their work from sponge-like systems, where the liquid is absorbed by the material under hydrostatic pressure, by noting that their mechanism leads to reversible cycles. This is correct, but it is not obvious to this reviewer that all previous sponge-like "negative" compressibility systems are irreversible (which I interpret as absorbing liquid under increased pressure, but not releasing liquid under reduced pressure). This might be true, but I feel that a more comprehensive review of the literature and a more explicit statement about this point is necessary here.

Even if there are reversible sponge-like systems, this work goes beyond in terms of exploring an underlying mechanism. Thus, even in that case, I believe the results in this paper merit publication in Nature Communications, as they will certainly appeal to the broad audience of this selective journal.

Reviewer #3 (Remarks to the Author):

The manuscript by Caprini et al. is a high-quality work proposing a straightforward strategy to obtain materials exhibiting the negative compressibility (NC) phenomenon exploiting capillary forces. The designed systems achieve the largest negative compressibility phenomenon reported so far and the physical principles underlying the presence of this effect in these systems are explained in a clear way.

While large negative compressibilities have been achieved in the past (Cairns, A.B.; Goodwin, A. L. *Phys. Chem. Chem. Phys.* 17, 20449–20465 (2015)), an intense research effort has been performed in recent times leading to outstanding works combining flexible, hydrophobic nanoporous materials and water (Tortora, M. et al. *Nano Letters* 21, 2848–2853 (2021); Zajdel, P. et al. *The Journal of Physical Chemistry Letters* 12, 4951–4957 (2021); Michel, L. et al. *Langmuir* 38, 211–220 (2022)). The present work is original in the sense that the metastable elastocapillary systems (MESs) concept introduced here explains its operation principles and provides a unified framework which applies to architectures extending from sub-nanometric porous materials to millimeter-scale hydrophobic metamaterials and nanometer-scale biological matter. Moreover, the measured negative compressibility in the millimeter-sized hydrophobic metamaterial consisting of pairs of Teflon facing

laminae, displays a huge NC of the order of -107 TPa^{-1} , which exceeds substantially the previous largest NC found in ZIF-8 (Tortora, M. et al. Nano Letters 21, 2848–2853 (2021)). For a similar ZIF material, ZIF-67, the construction by the authors, leads to a large negative compressibility slightly smaller than that of ZIF-8. Future designs constructed according to these principles could lead to new metamaterials with enhanced negative compressibilities and unforeseen material susceptibilities.

In the paper the information given is quite detailed so that the results are reproducible. This study deserves high visibility since it is a step forward in the NC research. Its relevance results from the important potential technological applications of the NC materials such as the development of ultrasensitive pressure-sensing devices, pressure-driven actuators, deep ocean optical telecommunication cables, artificial muscles, body armor, and devices for sound attenuation, superconductivity modulation, and transmission stabilization. Therefore, this manuscript, aside from being important in Materials Science, may also be important in solid state and theoretical physics, acoustics, telecommunications engineering, biophysics and many other branches of science.

However, I recommend a minor revision of the paper because there are some important issues that should be solved before the manuscript can be considered for publication in Nature. Therefore, I recommend a minor revision. I think that the following comments should be considered prior its publication.

1. In the first sentence of the introduction, the word “transition” should be replaced by “phenomenon”, since a large series of materials display NC without experiencing any transition.
2. This comment is very important and has a conceptual character. The definition of the negative compressibility (NC) given in the first sentence of the introduction is not correct: “NC is the ability of a material to respond to an external forcing by expanding in the direction of the force if the material is compressed or by contracting if decompressed”. The true definition of this effect is: “The ability of a material to respond to an external isotropic compression by expanding in one (linear) or two directions (area) or by contracting along one or two directions when decompressed”. That is, the external forcing must be uniform along all spatial directions (Cairns, A.B.; Goodwin, A. L. Phys. Chem. Chem. Phys. 17, 20449–20465 (2015); W. Miller, K.E. Evans, A. Marmier, Appl. Phys. Lett. 106, 231903 (2015)), and there is not a preferred direction along which the system expands (or contracts). This is not, in fact, a problem since, for example, the forces exerted over the crystalline material immersed in the liquid (Fig. 1) should be the same in all directions. However, the authors should correct this paragraph.
3. The replica of the graph showing the reversibility of the intrusion and extrusion cycles shown in Fig. 4.B. is very different to that given in Fig. 4.A. Is this large variability expected when replicating the other experimental designs performed in the paper?
4. Some additional recent references for applications of NC could be added: a) Superconductivity modulation (Uhoya, W., Tsoi, G., Vohra, Y.K., McGuire, M.A., Sefat, A.S., Sales, B.C., Mandrus, D., Weir, S.T., J. Phys Condens. Matter 22: 29220 (2010); b) Transmission stabilization; Jiang, X., Molokeyev, M.S., Dong, L., Dong, Z., Wang, N., Kang, L., Li, X., Li, Y., Tian, C., Peng, S., Li, W., Lin, Z, Nat. Commun. 11, 5593 (2020).

RESPONSE TO REVIEWERS' COMMENTS

In red Reviewers comments

In black authors responses

Reviewer #1

Authors describe a macroscopic laminae system of Teflon foils interacting with water under varied pressure conditions. There are several unclear points in this paper. The technical side of the experiment is not clear to me, and particularly the system change at point VI (page ??), between stages e-f in Figure 2. I would expect that when pressure is released water would stay between the films, unless the pressure reduced to vacuum – so all water evaporates. This makes the system operation really complicated. However, my main objection is the false use of term compressibility, which is very well defined in thermodynamics, as the negative volume change dV divided by (V multiplied by the pressure change dp). This definition must be applied for system under pressure, usually a uniform material. The reverse of compressibility is bulk modulus, characteristic of different materials. In the manuscript, Authors consider the volume of a sub-system in the bigger system under varied pressure. It should be made clear that the large-system volume is positively compressed, while the negative compressibility is claimed for the sub-system. It is not true, because its (sub-system) elements are not compressed at all – the films are pressurized from both inner and outer sides. Manuscript's SI video 2 showing a ball compressed in a liquid illustrates this misconception: in the left film the normal (positive compressibility) situation is shown, but the right film suggests that the negative compression is possible. But this right video neglects the intrusion of liquid (water) into the yellow ball. So the ball material transforms into another material. These serious mistakes invalidate this paper. There is no negative compressibility at all. For this reason I cannot recommend this paper for publication.

We thank the Reviewer for reading our manuscript. In the following we respond to the criticism raised, which seems mainly motivated by the concept and definition of compressibility that we used. The reviewer is right that compressibility is generally defined with reference to the entire system, nevertheless, in the literature cf. [Biochimica et Biophysica Acta 1386 (1998) 353-370, Nano Letters 21, 2848-2853 (2021)], and as agreed by the other two Reviewers below, this concept has been extended to sub-parts whose expansion can be triggered by some external stimulus. At first, the concept of compressibility of subsystems might seem at odds with the thermodynamics of uniform systems. Before delving into the technicalities of such a definition, we would like to remark why the present results are of interest for the community and for applications *independently* of the name one can give to the process. We consider *three* systems (new experiments: ZIF-67 and the laminae; from literature: biological ion channels) that all exhibit the following remarkable behaviour: by increasing the hydrostatic pressure, the solid part of the system expands more or less abruptly. This unusual behavior could be used in several

applications including pressure sensors, acoustic materials, pressure-driven actuators, artificial muscles, body armor, superconductivity modulation, and transmission stabilization (see also Rev. 3 comments). As an example, we report on the right figure a nanovalve application from [Tortora et al., *Nano Lett.* 2021, 21, 7, 2848], which shows how the behaviour of the solid part (as measured by its compressibility) is crucial for applications.

At least two of the considered systems also shrink while the pressure is reduced, due to the evaporation (ZIF-67) and to air bubble merging (laminae); in addition to making the operation of the system reversible, this behavior is also counterintuitive and can be harnessed in applications. The other merit of this work is to recognise the common elastocapillary origin of this behavior at very different scales and to provide the relevant theoretical framework. We hope to have convinced the Reviewer that, whether one calls this phenomenology “(negative) compressibility”, making connection to a specific field of research and to the recent literature [Refs. 1-9] or not the reported behaviour is quite unusual and interesting for applications, and the discovery of a general mechanism across ~9 orders of magnitude of length scales provides design principles to develop novel systems and interpret a broad phenomenology.

In the following we answer specific points.

Authors describe a macroscopic laminae system of Teflon foils interacting with water under varied pressure conditions.

The manuscript does not just present results on millimetre-sized laminae; rather, its strength is seamlessly joining new experiments on nanoporous materials (ZIF-67 *in operando* synchrotron experiments) with those on the millimeter scale (Teflon foils) through the simple yet powerful concept of metastable elastocapillary systems (MES).

The technical side of the experiment is not clear to me, and particularly the system change at point VI (page ??), between stages e-f in Figure 2. I would expect that when pressure is released water would stay between the films, unless the pressure reduced to vacuum – so all water evaporates.

Figure 2 reports the general theory for all the reported processes; point vi), corresponding to Fig. 2e-f, describes extrusion “which is possible only when the intrusion phenomenon is somewhat reversible”. For ZIF-67 (and for ion channels) local water evaporation (formation of water vapour bubbles) within the nanopores occurs even at very large hydrostatic pressures, while obviously liquid water is still present outside the porous grains. This local evaporation is due to the combination of extremely confined conditions with hydrophobicity; this phenomenology is well-known in nanoporous hydrophobic materials, see, e.g., [Eroshenko et al., *J. Am. Chem. Soc.* 123, 8129–8130 (2001)] and in some ion channels [Aryal et al. *J. Mol. Bio.* 427, 121–130 (2015)]. For the laminae, extrusion is *not* due to confined evaporation (formation of water vapour bubbles) but to the merging of pre-existing air bubbles (“Reversibility was surprisingly observed systems [48], although its origin is different: confinement-assisted nucleation of water vapour at the subnanoscale [28] and merging of air bubbles at the millimeter scale.”). More specifically “two asymmetric bubbles, remnants of the original air bridge, are still present on individual laminae” after intrusion, which “at pressures below the ambient one, [...] grow substantially, eventually merging again in an air bridge at $p = 0.985$ MPa”.

This makes the system operation really complicated.

The operation of the systems is not complicated because it consists only in increasing and decreasing hydrostatic pressure. It is certainly more involved to measure negative compressibility, i.e., demonstrating the principle as done in this work via synchrotron (ZIF-67) and an optically accessible pressurized cell (laminae), but this is not relevant for applications.

However, my main objection is the false use of term compressibility, which is very well defined in thermodynamics, as the negative volume change dV divided by (V multiplied by the pressure change dp). This definition must be applied for systems under pressure, usually a uniform material. The reverse of compressibility is bulk modulus, characteristic of different materials. In the manuscript, Authors consider the volume of a sub-system in the bigger system under varied pressure.

The point about compressibility is an important one which we want to clarify beyond doubt. We carefully stated the definition of compressibility adopted in the manuscript, which, as discussed at the beginning of this response, is consistent with the literature in the field. In our work, we are interested in the response of the solid (ZIF-67, proteins, or the laminae) cf. [Biochimica et Biophysica Acta 1386 (1998) 353-370, Nano Letters 21, 2848-2853 (2021)] under increasing/decreasing hydrostatic pressure. Compressibility, here, refers solely to the solid part of our system, which is also subjected to hydrostatic pressure. We consider this solid part as a black box and *measure* experimentally its volume changes while the driving force is changed. These observables are the relevant ones because the systems that we use (ZIF-67 and the laminae) have a controlled hydrostatic pressure and we can measure the volume of the solid. Regardless of what happens within the solid, therefore, the compressibility of this sub-system is well defined, although it does not coincide with the one of the entire system. The difference is well highlighted in the manuscript such that the reader cannot get confused.

In more theoretical and abstract terms, the thermodynamics of a system with different phases and interfaces is best discussed in the grand canonical ensemble [Evans et al., J. Chem. Phys. 84, 2376 (1986)]. In this framework, the chemical potential is constant across the system (liquid, vapor/gas, solid, interfaces); the possible misunderstanding when reasoning in terms of pressure, which is legitimate but insidious, comes from the fact that this observable, instead, can be different in the liquid (hydrostatic pressure) and in the vapour/gas phase (vapour/gas pressure). Within the grand canonical framework, it is immediately clear that our experiments consist in first increasing the chemical potential, thus triggering intrusion, and subsequently decreasing it, to cause extrusion. These transitions from a metastable to a stable state result in a negative susceptibility (exotic response of the system in the same direction of the applied thermodynamic force). However, we are forced to use the pressure as a variable (and the related concept of compressibility) because this is the experimentally accessible variable; it is however possible to relate the hydrostatic pressure to the chemical potential by a first order expansion of pressure in terms of the chemical potential in the vicinity of coexistence conditions, which yields $\mu - \mu_{\text{coex}} \sim \rho_L p_L$ with ρ_L and p_L the liquid number density and pressure, respectively [Evans et al., J. Chem. Phys. 84, 2376 (1986)].

It should be made clear that the large-system volume is positively compressed, while the negative compressibility is claimed for the sub-system.

We agree with the Reviewer that it should be mentioned that the overall volume of the system shrinks upon intrusion. Indeed, we intended to make it as clear as

possible in the original version of the manuscript, see Fig. 1A, and by several clear explanations including photos of the experimental setup and video demonstrations. We have added the sentence: “We remark that, even if the solid part of MESs presents substantial NC, the overall system, which includes the liquid, does not exhibit such feature, see Fig. 1A”.

It is not true, because its (sub-system) elements are not compressed at all – the films are pressurized from both inner and outer sides.

The crucial point is that the thermodynamic state of our (composite) system is defined by its hydrostatic pressure because this is the control variable. The presence of complex interfaces, of different phases (gaseous), and of elastic solids is not in contrast with the notion of a system under constant pressure and does not impede to define its compressibility (see also the equivalent discussion in terms of chemical potential above). This control variable is the driving force of the processes under scrutiny: intrusion and extrusion. The Reviewer is right that, microscopically (within internal degrees of freedom of the thermodynamic system), the unexpected expansion behaviour of MESs is due to the crucial difference between having a gas inside the cavities or a liquid; but this is a phase transition which is triggered by the change of the control parameter, i.e., the hydrostatic pressure (or chemical potential). The thermodynamics of this system and its metastable states is rigorously described by Eq. (1) and is the key subject of the present manuscript, which was never discussed before in the literature. Indeed, this is the concept of MES: exploiting metastabilities (phase change from confined vapour/gas to confined liquid mediated by capillarity) to trigger a compression-induced expansion and a decompression-induced shrinking.

Manuscript's SI video 2 showing a ball compressed in a liquid illustrates this misconception: in the left film the normal (positive compressibility) situation is shown, but the right film suggests that the negative compression is possible. But this right video neglects the intrusion of liquid (water) into the yellow ball.

We concur with the Reviewer that the video was too simplified and did not clearly show what happens within the pore; the correct visualization is that in Fig. 1A, according to which the video has been amended to the best of our capabilities. A new version of the video is uploaded. However, this visualization issue does not alter the main claim: the volume of the solid subsystem does expand when the pressure is increased; there is no transformation of the material, specifically chemical transformation, upon intrusion. The solid material is always chemically and physically distinct from the wetting liquid and the process is reversible.

So the ball material transforms into another material.

No, it does not. We changed the supplementary video to solve any graphical issue, and we hope we clarified in the previous comment.

These serious mistakes invalidate this paper. There is no negative compressibility at all. For this reason I cannot recommend this paper for publication.

For the arguments exposed above, we disagree with the conclusions of the Reviewer, because i) we transparently defined the compressibility of the solid subsystem according to the recent literature on the subject; ii) this subsystem does undergo expansion when compressed and shrinking when decompressed, leading to negative compressibility according to the adopted definition; iii) there is no conflict with classical thermodynamics, because the system is metastable *by construction*; iv) the phase transitions of the systems

are driven by the control parameter, i.e., the hydrostatic pressure – the rigorous thermodynamics of this elastocapillary system is the subject of this work. Finally, we remark that the reported MES behaviour is exotic, cross-field, and cross-scale and lends itself to new applications, independently of the special definition adopted for (negative) compressibility; the presented framework provides general principles to design other systems with analogous characteristics.

Reviewer #2

The paper “Metastable elastocapillary systems with negative compressibility” offers exciting new results on how to generate negative compressibility in metastable systems. Negative compressibility, as precisely defined in this manuscript, cannot occur continuously in thermodynamical equilibrium but has been previously shown to occur through phase transitions. In that case, the system transitions from one stable/metastable state to another in response to increased stress.

Here the authors explore another mechanism that can lead to negative compressibility, which is nicely illustrated in Fig 2a. The mechanism involves intrusion and extrusion of fluid particles under hydrodynamical compression/decompression. As pressure is increased, intrusion causes the material to expand.

The paper is very well written, with a good balance of experiments and numerical + analytical description. The paper is technically correct, and the results support the conclusions. The figures are clear, and the literature is suitably covered. Supplemental materials are relevant and should be published with the paper.

We thank the Reviewer for the very positive feedback on the manuscript. In the following we respond to the main points raised.

I recommend publication in Nature Communications, provided the authors can address the following minor point in their revisions:

In p. 2, the authors appear to distinguish the mechanism in their work from sponge-like systems, where the liquid is absorbed by the material under hydrostatic pressure, by noting that their mechanism leads to reversible cycles. This is correct, but it is not obvious to this reviewer that all previous sponge-like "negative" compressibility systems are irreversible (which I interpret as absorbing liquid under increased pressure, but not releasing liquid under reduced pressure). This might be true, but I feel that a more comprehensive review of the literature and a more explicit statement about this point is necessary here.

As far as we understood, the pressure-induced expansion in the cited sponge-like systems are not reversible (at least not completely). However, we restructured the paragraph in the introduction to clearly distinguish the case of sponges from that of hydrophilic zeolites undergoing high pressure changes in the matrix by the formation of hydrates. These two cases were initially mentioned together only because of the chemical changes in the matrix, which is excluded in our systems; the focus was not specific to “reversibility”, but rather to chemical changes. In the work by Lee et al. (Zeolites, Natrolite Family, JACS 2002), which deals with zeolites, it is stated that the “expanded phase, stable at high pressure, is retained when the pressure is released”. The full paragraph was rearranged for the sake of clarity and to highlight the main point (absence of chemical changes), now reading:

“In addition, at a variance with the continuous water uptake at the origin of sponge swelling, which alters the chemical interactions between cellulose nanofibrils or nanocrystals [27–29], the NC process discussed in this work is purely mechanical, corresponding to a first order transition between a stable and a metastable state, i.e., confined liquid and confined vapour, while the chemistry of the solid medium is not altered. The reported NC mechanisms rely on reversible physical processes occurring at low pressures. In contrast, the pressure-induced formation of an expanded (super)hydrated phase

in zeolites [27] was reported at high pressures (GPa); this transformation was retained after pressure release. Similarly, in sponge-like cellulose systems the swelling is associated to a more complex process, starting with a chemical interaction between water and cellulose hydroxyl groups, resulting into a partially irreversible structure modification of the system [28, 29], that cannot be restored simply by releasing the pressure, but requires complete drying of the matrix.”

Even if there are reversible sponge-like systems, this work goes beyond in terms of exploring an underlying mechanism. Thus, even in that case, I believe the results in this paper merit publication in Nature Communications, as they will certainly appeal to the broad audience of this selective journal.

We thank the Reviewer once again for recommending the publication of this manuscript in Nature Communications.

Reviewer #3

The manuscript by Caprini et al. is a high-quality work proposing a straightforward strategy to obtain materials exhibiting the negative compressibility (NC) phenomenon exploiting capillary forces. The designed systems achieve the largest negative compressibility phenomenon reported so far and the physical principles underlying the presence of this effect in these systems are explained in a clear way.

While large negative compressibilities have been achieved in the past (Cairns, A.B.; Goodwin, A. L. *Phys. Chem. Chem. Phys.* 17, 20449–20465 (2015)), an intense research effort has been performed in recent times leading to outstanding works combining flexible, hydrophobic nanoporous materials and water (Tortora, M. et al. *Nano Letters* 21, 2848–2853 (2021); Zajdel, P. et al. *The Journal of Physical Chemistry Letters* 12, 4951–4957 (2021); Michel, L. et al. *Langmuir* 38, 211–220 (2022)). The present work is original in the sense that the metastable elastocapillary systems (MESs) concept introduced here explains its operation principles and provides a unified framework which applies to architectures extending from sub-nanometric porous materials to millimeter-scale hydrophobic metamaterials and nanometer-scale biological matter. Moreover, the measured negative compressibility in the millimeter-sized hydrophobic metamaterial consisting of pairs of Teflon facing laminae, displays a huge NC of the order of -107 TPa^{-1} , which exceeds substantially the previous largest NC found in ZIF-8 (Tortora, M. et al. *Nano Letters* 21, 2848–2853 (2021)). For a similar ZIF material, ZIF-67, the construction by the authors, leads to a large negative compressibility slightly smaller than that of ZIF-8. Future designs constructed according to these principles could lead to new metamaterials with enhanced negative compressibilities and unforeseen material susceptibilities.

In the paper the information given is quite detailed so that the results are reproducible. This study deserves high visibility since is a step forward in the NC research. Its relevance results from the important potential technological applications of the NC materials such as the development of ultrasensitive pressure-sensing devices, pressure-driven actuators, deep ocean optical telecommunication cables, artificial muscles, body armor, and devices for sound attenuation, superconductivity modulation, and transmission stabilization. Therefore, this manuscript, aside from being important in Materials Science, may also be important in solid state and theoretical physics, acoustics, telecommunications engineering, biophysics and many other branches of science.

We thank the Reviewer for positively assessing this work and for recognising its relevance for potential technological applications.

However, I recommend a minor revision of the paper because there are some important issues that should be solved before the manuscript can be considered for publication in Nature. Therefore, I recommend a minor revision. I think that the following comments should be considered prior its publication.

1. In the first sentence of the introduction, the word “transition” should be replaced by “phenomenon”, since a large series of materials display NC without experiencing any transition.

We changed the wording according to this suggestion.

2. This comment is very important and has a conceptual character. The definition of the negative compressibility (NC) given in the first sentence of the introduction is not correct: “NC is the ability of a material to respond to an external forcing by expanding in the direction of the force if the material is compressed or by contracting if decompressed”. The true definition of this effect is: “The ability of a material to respond to an external isotropic compression by expanding in one (linear) or two directions (area) or by contracting along one or two directions when decompressed”. That is, the external forcing must be uniform along all spatial directions (Cairns, A.B.; Goodwin, A. L. Phys. Chem. Chem. Phys. 17, 20449–20465 (2015); W. Miller, K.E. Evans, A. Marmier, Appl. Phys. Lett. 106, 231903 (2015)), and there is not a preferred direction along which the system expands (or contracts). This is not, in fact, a problem since, for example, the forces exerted over the crystalline material immersed in the liquid (Fig. 1) should be the same in all directions. However, the authors should correct this paragraph.

We thank the Reviewer for this comment. The suggested phrasing describes exactly negative linear and area compressibilities, which however are not the sole subject of our work. We also report on negative volume compressibility (ZIF-67), so we have to provide a more general definition in the first sentence. The amended version now reads: “NC relates to the ability of a material to respond to an external isotropic compression by expanding or by contracting when decompressed”.

3. The replica of the graph showing the reversibility of the intrusion and extrusion cycles shown in Fig. 4.B. is very different to that given in Fig. 4.A. Is this large variability expected when replicating the other experimental designs performed in the paper?

The Reviewer is right that the experiments on laminae present some variability between replicas in Fig. S4, which originates in i) the pinning phenomena due to fine details of the lamina surface and geometry and ii) the initial conditions of the bubble. In the future, one could mitigate both issues by more careful designs of hydrophobic laminae or automated manufacturing and experimental procedures. For ZIF-67, the negative compressibility phenomenology is robust over different replicas, showing good reproducibility (see also the results obtained for three different ZIF-8 samples [A, B] and on the Cu₂(tebpz) MOF [C]). However, we note that, although the crystal structure is always the same, several details may change from batch to batch, such as size of the grains, number of defects, etc. which might lead to quantitative differences in the cycles, i.e., variability. In particular, we have recently demonstrated how crystallite size affects the negative compressibility of ZIF-8 [B]. It should be noted that water intrusion-extrusion into-from ZIF-67 (by liquid porosimetry, not characterizing negative compressibility, which requires synchrotron/neutron scattering experiments) has been explored extensively by different groups demonstrating very similar behaviour [D, E]. Certainly, there are some variabilities in the intrusion/extrusion pressures. However, we recently demonstrated that this predominantly depends on the synthesis protocol and can be avoided by carefully controlling sample quality [F]. In summary, the focus of this first work is to ensure the robustness of the phenomenology, i.e., its repeatability, rather than engineering its sample-to-sample variability, which will be done next when a specific application and the related constraints will be defined; this was clarified by

adding the following sentence to the Discussion: “For specific applications, the MES properties and their variability could be tailored by carefully controlling sample quality [52] (subnanoMES) or automated manufacturing and experimental procedures (milliMES).”.

[A] Tortora, M., Zajdel, P., Lowe, A.R., Chorażewski, M., Leão, J.B., Jensen, G.V., Bleuel, M., Giacomello, A., Casciola, C.M., Meloni, S. and Grosu, Y., 2021. Giant negative compressibility by liquid intrusion into superhydrophobic flexible nanoporous frameworks. *Nano Letters*, 21(7), pp.2848-2853.

[B] Johnson, L.J., Mirani, D., Le Donne, A., Bartolomé, L., Amayuelas, E., López, G.A., Grancini, G., Carter, M., Yakovenko, A.A., Trump, B.A. and Meloni, S., 2023. Effect of Crystallite Size on the Flexibility and Negative Compressibility of Hydrophobic Metal–Organic Frameworks. *Nano Letters*. 23, 23, 10682–10686

[C] Zajdel, P., Chorażewski, M., Leao, J.B., Jensen, G.V., Bleuel, M., Zhang, H.F., Feng, T., Luo, D., Li, M., Lowe, A.R. and Geppert-Rybczynska, M., 2021. Inflation Negative Compressibility during Intrusion–Extrusion of a Non-Wetting Liquid into a Flexible Nanoporous Framework. *The Journal of Physical Chemistry Letters*, 12(20), pp.4951-4957.

[D] Khay, I., Chaplais, G., Nouali, H., Ortiz, G., Marichal, C. and Patarin, J., 2016. Assessment of the energetic performances of various ZIFs with SOD or RHO topology using high-pressure water intrusion–extrusion experiments. *Dalton Transactions*, 45(10), pp.4392-4400.

[E] Grosu, Y., Gomes, S., Renaudin, G., Grolier, J.P.E., Eroshenko, V. and Nedelec, J.M., 2015. Stability of zeolitic imidazolate frameworks: effect of forced water intrusion and framework flexibility dynamics. *RSC advances*, 5(109), pp.89498-89502.

[F] Amayuelas, E., Bartolomé, L., Zhang, Y., Lopez del Amo, JM., Bondarchuk, O., Nikulin, A., Bonilla, F., Palomo del Barrio, E., Zajdel, P. and Grosu, Y., Quality-Dependent Performance of Hydrophobic Zif-67 Upon High-Pressure Water Intrusion-Extrusion Process. *Phys Chem Chem Phys*. Accepted. 10.1039/D3CP03519K 2023

4. Some additional recent references for applications of NC could be added: a) Superconductivity modulation (Uhoya, W., Tsoi, G., Vohra, Y.K., McGuire, M.A., Sefat, A.S., Sales, B.C., Mandrus, D., Weir, S.T., J. Phys Condens. Matter 22: 29220 (2010); b) Transmission stabilization; Jiang, X., Molochev, M.S., Dong, L., Dong, Z., Wang, N., Kang, L., Li, X., Li, Y., Tian, C., Peng, S., Li, W., Lin, Z, Nat. Commun. 11, 5593 (2020).

We thank the Reviewer for these additional interesting references, which have been added to the first paragraph of the introduction.

REVIEWER COMMENTS

Reviewer #1 (Remarks to the Author):

Report on the revision of manuscript:
Metastable elastocapillary systems with negative compressibility
by Davide Caprini et al.

In the revision, Authors corrected some of the errors indicated in my previous report. Their new version of Movie S1 goes in the right direction, although it illustrates that the described transformation is complex, as opposed to the simple nature of the concept of compressibility. After watching this movie the Reader will immediately question term 'compressibility' used in the title and text.

On the other hand, Authors insist on describing their system transformations as 'compressibility'. One of their main arguments is the precedents in using the term 'compressibility'. They respond: In our work, we are interested in the response of the solid (ZIF-67, proteins, or the laminae) cf. [Biochimica et Biophysica Acta 1386 (1998) 353-370, Nano Letters 21, 2848-2853 (2021)] under increasing/decreasing hydrostatic pressure. Compressibility, here, refers solely to the solid part of our system, which is also subjected to hydrostatic pressure. We consider this solid part as a black box and measure experimentally its volume changes while the driving force is changed. These observables are the relevant ones because the systems that we use (ZIF-67 and the laminae) have a controlled hydrostatic pressure and we can measure the volume of the solid. Regardless of what happens within the solid, therefore, the compressibility of this sub-system is well defined, although it does not coincide with the one of the entire system. The difference is well highlighted in the manuscript such that the reader cannot get confused.

The first of the indicated papers [Biochimica et Biophysica Acta 1386 (1998) 353-370] is a review, where in Table 1 in one instant a negative compressibility is given for the molecule of amino acid diluted solution (as explained by the footnote to that entry). Can we call a molecule a SOLID component of a diluted solution? Moreover, while the volume of a molecule in different environments is still disputable, authors Heremans and Smeller indicated that the increased volume of the amino acid molecules is only a possible explanation of the ultra-sound velocity measurements performed on the diluted solutions of amino acids under moderate pressure. Without discussing here the validity of this assumption (if unfolded chains are more voluminous than folded ones, etc.), in my opinion, this argument is truly far away from the compressibility of solids.

The second indicated paper originated from the same group of authors as the reviewed manuscript, including Zajdel, Giacomello, Casciola, Meloni and Grosu. In that paper they describe a typical event of the sorption increasing the volume of crystals – they call this 'negative compressibility', which is wrong according to the physical definition: the transformation described in the paper is neither monotonic, nor single-compound, nor even a closed system! At the same time, apart from their own papers with wrong definitions, Authors cite the papers with the correct approach – which describe the analogous transformations as deformations (e.g. ref. 21 Michel et al.) or as in ref. 22 (Krause et al.). The authors of this latter paper carefully only once in all text compare the described effect to compressibility by writing [quote]: 'may be interpreted as an adsorptive analogue of force-amplifying negative compressibility'. They carefully respect the definitions, consistency and clarity of presentation.

I conclude that the revision will further confuse the meaning of the well-defined term 'compressibility' and therefore I cannot recommend its publication.

Reviewer #2 (Remarks to the Author):

I've read the revised manuscript as well as the authors' responses to all three reviewers. The authors satisfactorily addressed the crucial question raised in my report on the distinction between this work and previous work involving intrusion. The revised manuscript represents a substantial advance in the study of elastocapillary in material design.

However, I'm afraid I have to disagree with some changes that the authors were asked to implement to accommodate the requests of reviewer #3. The effect reported in this manuscript is a genuine form of negative compressibility, where the change in the volume ΔV of the elastocapillary system is the opposite of the usual one in response to a change in ΔP . Here the authors are very careful to both i) treat pressure as the independent variable and ii) to measure expansion in volume (rather than overcompensation of expansion in one direction by contraction in others). As clear as these two requirements should be, they are violated in most papers that reviewer #3 requested to be cited. Even though those papers use the words "negative compressibility" in a lax way, the effects they report are weaker than *the negative of the quantity defined as compressibility*. This is the case for the excellent reason that true negative (volumetric) compressibility cannot occur continuously in isolated systems (it is thermodynamically forbidden). Still, it can occur through transitions (as shown in previous work) and in mechanically open systems (as shown in this work).

The definition of reviewer #3 is an attempt to accommodate as negative compressibility situations where the material expands in one direction in response to pressure increase, even though it contracts along the other directions and in volume (and thus the material may have positive compressibility in most directions, across all planes, and in volume). The definition of negative compressibility proposed by reviewer #3 is not really a definition but rather a nickname adopted by some authors to highlight one aspect of their materials that had some resemblance to one aspect of negative compressibility. In this sense, even though the added references can help contextualize the work, the change in the definition requested by reviewer #3 is a conceptual error.

That said, the overall presentation in the manuscript is rigorous and clear. It is consistent with the best conventions in the field. I recommend publication in Nature Communications.

Reviewer #3 (Remarks to the Author):

The revised version of the manuscript "Metastable elastocapillary systems with negative compressibility" by Caprini et al. has been largely improved with respect to the original version of the manuscript, and all the reviewers' comments have been responded properly. Therefore, I recommend publication of the manuscript in Nature. However, the following comments should be considered prior to the final publication.

1. As stated in the response to the first reviewer, compressibility is generally defined with reference to the entire system. Therefore, the fact that the compressibility concept used here in the manuscript refers to a subsystem should be clearly noted and explained in the introduction of the manuscript.

2. In video S1. Please maintain the legends "Positive compressibility", "Negative compressibility" and the corresponding equations for beta (β) in either case.

3. The issue concerning the repeatability of the intrusion and extrusion cycles is important. Thus, in page 5, the first sentence of the second column "... when the pressure is increased, closing the cycle, which is largely reversible and repeatable (Fig. S4).", should be supplemented with a pair of additional sentences such as: "It should be noted that the reversibility of intrusion and extrusion cycles presents some variability (Fig. S4.B), originated from: i) the pinning phenomena due to fine details of the lamina surface and geometry, and ii) the initial conditions of the bubble. This variability could be mitigated by using more careful designs of hydrophobic laminae or automated manufacturing and experimental procedures" (apart from the sentence added in the Discussion: "For specific applications, the MES properties and their variability could be tailored by carefully controlling sample quality [52] (subnanoMES) or automated manufacturing and experimental procedures (milliMES).", which should be in the text).

RESPONSE TO REVIEWERS' COMMENTS

Reviewers comments are in black

Authors response are in red

Reviewer #1 (Remarks to the Author):

Report on the revision of manuscript:

Metastable elastocapillary systems with negative compressibility

by Davide Caprini et al.

In the revision, Authors corrected some of the errors indicated in my previous report. Their new version of Movie S1 goes in the right direction, although it illustrates that the described transformation is complex, as opposed to the simple nature of the concept of compressibility. After watching this movie the Reader will immediately question term 'compressibility' used in the title and text.

On the other hand, Authors insist on describing their system transformations as 'compressibility'. One of their main arguments is the precedents in using the term 'compressibility'. They respond:

In our work, we are interested in the response of the solid (ZIF-67, proteins, or the laminae) cf. [Biochimica et Biophysica Acta 1386 (1998) 353-370, Nano Letters 21, 2848-2853 (2021)] under increasing/decreasing hydrostatic pressure. Compressibility, here, refers solely to the solid part of our system, which is also subjected to hydrostatic pressure. We consider this solid part as a black box and measure experimentally its volume changes while the driving force is changed. These observables are the relevant ones because the systems that we use (ZIF-67 and the laminae) have a controlled hydrostatic pressure and we can measure the volume of the solid. Regardless of what happens within the solid, therefore, the compressibility of this sub-system is well defined, although it does not coincide with the one of the entire system. The difference is well highlighted in the manuscript such that the reader cannot get confused.

The first of the indicated papers [Biochimica et Biophysica Acta 1386 (1998) 353-370] is a review, where in Table 1 in one instant a negative compressibility is given for the molecule of amino acid diluted solution (as explained by the footnote to that entry). Can we call a molecule a SOLID component of a diluted solution? Moreover, while the volume of a molecule in different environments is still disputable, authors Heremans and Smeller indicated that the increased volume of the amino acid molecules is only a possible explanation of the ultra-sound velocity measurements performed on the diluted solutions of amino acids under moderate pressure. Without discussing here the validity of this assumption (if unfolded chains are more voluminous than folded ones, etc.), in my opinion, this argument is truly far away from the compressibility of solids.

The second indicated paper originated from the same group of authors as the reviewed manuscript, including Zajdel, Giacomello, Casciola, Meloni and Grosu. In that paper they describe a typical event of the sorption increasing the volume of crystals – they call this

'negative compressibility', which is wrong according to the physical definition: the transformation described in the paper is neither monotonic, nor single-compound, nor even a closed system! At the same time, apart from their own papers with wrong definitions, Authors cite the papers with the correct approach – which describe the analogous transformations as deformations (e.g. ref. 21 Michel et al.) or as in ref. 22 (Krause et al.). The authors of this latter paper carefully only once in all text compare the described effect to compressibility by writing [quote]: 'may be interpreted as an adsorptive analogue of force-amplifying negative compressibility'. They carefully respect the definitions, consistency and clarity of presentation. I conclude that the revision will further confuse the meaning of the well-defined term 'compressibility' and therefore I cannot recommend its publication.

Reviewer 1 asserts that this work and many others are misusing the term “compressibility”, suggesting that this word can be only applied to “monotonic transformations, [...], single-compound, closed systems”. This definition is reductive and somewhat arbitrary. Indeed, the terms “compressibility” is defined as one of the most common “response function” of a general “thermodynamic system”:

*“The response functions are the thermodynamic quantities most accessible to experiment. They give us information about how a specific state variable changes as other independent state variables are changed under controlled conditions [...]. The response functions can be divided into (a) thermal response functions, such as heat capacities, (b) **mechanical response functions**, such as **compressibility** and **susceptibility**, and (c) chemical response functions.”*

from **A Modern Course in Statistical Physics, 4ed, L.E. Reichl.**

To the best of our knowledge, the concept of “thermodynamic system” is broader than simple systems composed by “single-compound”, “closed systems” undergoing “monotonic transformations”, see, e.g.:

1) *“The macroscopic physical object of interest delimited by real or imaginary boundaries is referred to as an assembly, and that which is external to the boundary is referred to as the surroundings. The assembly is itself composed of atoms, molecules, electrons, ions, etc., but the thermodynamics is blind to these.”* from **Thermodynamics and Statistical Mechanics, L. M. Grossman, 1969.**

2) *A thermodynamic system is a quantity of matter of fixed identity, around which we can draw a boundary. The boundaries may be fixed or moveable. Work or heat can be transferred across the system boundary. Everything outside the boundary is the surroundings.”* from **Thermodynamics and Propulsion, Spakovszky, MIT Course 16 Fall 2002**

3) *“A thermodynamic system is a macroscopic system. Thermodynamics always divides the universe into the system and its surroundings. [...] The fundamental parameters that define a thermodynamic state, such as the pressure P , volume V , the temperature T , and the total mass M or number of moles n are measurable quantities assumed to be provided experimentally.”* from **Statistical Mechanics: Theory and Molecular Simulation, M.E. Tuckerman**

In none of these books we have found restrictions to the application of the definitions of “system” and “compressibility” to “single compound, closed systems”, or other restrictions (“solids”) mentioned in the Reviewer comment. For example, compressibility is routinely

used in the grand canonical ensemble, which is an open system, to evaluate fluctuations in particle number [Tuckerman, *ibidem*, Ch. 6].

Indeed, also looking at other references, it seems that the most general definition of “compressibility” is simply the derivative of the volume of such “thermodynamic system” or “assembly” or “object” with respect to pressure, at constant temperature or constant entropy – which is the one we compute in our experiments:

“The compressibility is the rate of change of volume with pressure, and this can be done at constant temperature or constant entropy.” from **Thermodynamics and Statistical Mechanics, P. Attard, 2002**

We apply this definition to the proposed class of “metastable elastocapillary systems”, reporting the theoretical model explaining our concept (Eq. 1) and two experimental measurements on millimetre and nanometre systems.

Finally, we agree that the compressibility can have different phenomenology for different systems and ensembles. Indeed, we agree that protein compressibility (soft matter) is “truly far away from the compressibility of solids”. Nevertheless, there is no reason why this difference should make the term “compressibility”, **measured as the volume changes in response to pressure changes**, only referable to solids (“... truly far away from the compressibility of solids”) and thus invalid for the “soft” case. Unfortunately, the Reviewer did not give us any reference supporting “the physical definition”, “well-defined” compressibility she/he is referring to. Hence, without knowing the Reviewer’s background nor other references, it is impossible for us to understand where the criticisms lie and make constructive modifications to our manuscript inspired by her/his comments, despite the fact that we are confident that we are using the term in a correct way.

In conclusion, while we further modified the text to make clear the definition of compressibility, we could not constructively address the specific comment of the Reviewer because the relevant references and definitions were missing.

Reviewer #2 (Remarks to the Author):

I've read the revised manuscript as well as the authors' responses to all three reviewers. The authors satisfactorily addressed the crucial question raised in my report on the distinction between this work and previous work involving intrusion. The revised manuscript represents a substantial advance in the study of elastocapillary in material design.

We thank the Reviewer for the appreciation for our work.

However, I'm afraid I have to disagree with some changes that the authors were asked to implement to accommodate the requests of reviewer #3. The effect reported in this manuscript is a genuine form of negative compressibility, where the change in the volume ΔV of the elastocapillary system is the opposite of the usual one in response to a change in ΔP . Here the authors are very careful to both i) treat pressure as the independent variable and ii) to measure expansion in volume (rather than overcompensation of expansion in one direction by contraction in others). As clear as these two requirements should be, they are violated in most papers that reviewer #3 requested to be cited. Even though those papers use the words "negative compressibility" in a lax way, the effects they report are weaker than *the negative of the quantity defined as compressibility*. This is the case for the excellent reason that true negative (volumetric) compressibility cannot occur continuously in isolated systems (it is thermodynamically forbidden). Still, it can occur through transitions (as shown in previous work) and in mechanically open systems (as shown in this work). The definition of reviewer #3 is an attempt to accommodate as negative compressibility situations where the material expands in one direction in response to pressure increase, even though it contracts along the other directions and in volume (and thus the material may have positive compressibility in most directions, across all planes, and in volume). The definition of negative compressibility proposed by reviewer #3 is not really a definition but rather a nickname adopted by some authors to highlight one aspect of their materials that had some resemblance to one aspect of negative compressibility. In this sense, even though the added references can help contextualise the work, the change in the definition requested by reviewer #3 is a conceptual error.

We thank the Reviewer for the thorough analysis and for providing us with the opportunity to make the definition of negative compressibility more rigorous in the manuscript: we recognize that the clear explanation given by the Reviewer is the most general and thermodynamic consistent one, avoiding confusion with other phenomena such as negative Poisson ratios. We tried to use it consistently at all points in the text, avoiding the term "linear negative compressibility" also when referring to expansion/contraction measured in a single direction. On the other hand we also agree with the Reviewer that a broad span of references is helpful to "contextualise the work".

In the first paragraph of the Introduction, we gave a broad perspective on the possible (mis)uses of the term negative compressibility:

"Negative compressibility (NC) is a term used in literature to broadly indicate the unusual behaviour of materials expanding when compressed or contracting when decompressed. The term has been adopted for linear [1–4], area [1, 5, 6], or volume [7, 8] NC."

While in the second paragraph we provided the exact definition used in this work (following the definition provided by the Reviewer):

*“A careful definition of NC is needed to distinguish cases in which the expansion upon compression in one direction is overcompensated by contraction in the other directions from those in which the volume of the system undergoes expansion [7], i.e., **the negative of the quantity defined as compressibility** -- the definition used in this work. This is the most stringent case because, to ensure mechanical stability, thermodynamics forbids this kind of NC for closed systems in thermodynamic equilibrium (i.e., when the system is equilibrated for unlimited time) [19, 20] as anticipated in the context of the Landau theory of phase transformations, see also [21]. In this work and in others [7], true NC has been achieved by lifting the hypotheses on i) on the kind of system or on ii) thermodynamic equilibrium by considering open systems or metastable states, respectively. Notable examples demonstrating the existence of true NC are metamaterials exhibiting transitions between metastable solid phases [20] and the metastable elastocapillary systems (MESS) [8] considered here, **whose solid component is an open system exhibiting NC.**”*

We are confident that this solution allows for a rigorous definition of NC while keeping a broad perspective on the literature/context.

That said, the overall presentation in the manuscript is rigorous and clear. It is consistent with the best conventions in the field. I recommend publication in Nature Communications.

We thank Reviewer 2 again for supporting our work.

Reviewer #3 (Remarks to the Author):

The revised version of the manuscript “Metastable elastocapillary systems with negative compressibility” by Caprini et al. has been largely improved with respect to the original version of the manuscript, and all the reviewers’s comments has been responded properly. Therefore, I recommend publication of the manuscript in Nature.

We thank the Reviewer for his/her previous comments that helped us to improve the manuscript, and the appreciation of the final result.

However, the following comments should be considered prior to the final publication.

1. As stated in the response to the first reviewer, compressibility is generally defined with reference to the entire system. Therefore, the fact that the compressibility concept used here in the manuscript refers to a subsystem should be clearly noted and explained in the introduction of the manuscript.

We have better clarified that our definition of compressibility is always referred to the solid material, in the **Abstract**:

“the solid part of such metastable elastocapillary systems displays negative compressibility across different scales: hydrophobic microporous materials, proteins, and millimetre-sized laminae.

and the **Introduction**:

“Notable examples demonstrating the existence of true NC are metamaterials exhibiting transitions between metastable solid phases [7, 20] and the metastable elastocapillary systems (MESs) considered here, whose solid component is an open system exhibiting NC. MESs have been realised by using flexible, hydrophobic nanoporous materials immersed in water [8, 22]”

We note that the solid part might be viewed either as a subsystem of the closed heterogeneous system or as an open system as suggested by Reviewer #2. We adopted this latter definition for consistency, but this choice does not limit the generality.

We further point out that in the previous version it was already stated (page 3, left column):

“[...] even if the solid part of MESs presents substantial NC, the overall system, which includes the liquid, does not exhibit such feature, see Fig. 1A.”

2. In video S1. Please maintain the legends “Positive compressibility”, “Negative compressibility” and the corresponding equations for beta (β) in either case.

We have updated the video according to the suggestion, and also added the sentence “Negative compressibility is displayed by the solid part of MESs.” to Fig. 1A that, together with Fig. 2A, illustrates beyond doubt what is the “subsystem” undergoing expansion.

3. The issue concerning the repeatability of the intrusion and extrusion cycles is important. Thus, in page 5, the first sentence of the second column “... when the pressure is increased, closing the cycle, which is largely reversible and repeatable (Fig. S4).”, should be

supplemented with a pair of additional sentences such as: “It should be noted that the reversibility of intrusion and extrusion cycles presents some variability (Fig. S4.B), originated from: i) the pinning phenomena due to fine details of the lamina surface and geometry, and ii) the initial conditions of the bubble. This variability could be mitigated by using more careful designs of hydrophobic laminae or automated manufacturing and experimental procedures” (apart from the sentence added in the Discussion: “For specific applications, the MES properties and their variability could be tailored by carefully controlling sample quality [52] (subnanoMES) or automated manufacturing and experimental procedures (milliMES).”, which should be in the text).

We thank the Review for highlighting this point. We highlight that the variability of the intrusion/extrusion cycle can be reduced with a more careful design of the laminae and by modifying the experimental procedures aiming at reducing/avoiding the presence of undesirable air bubbles in specific parts of the system. We modified the text according to the Referee’s indication:

“We highlight that the intrusion and extrusion cycles present some variability among independent replicas (Fig. S4B), originating in i) the pinning phenomena due to fine details of the lamina surface and geometry, and ii) the initial conditions of the bubble. This variability could be mitigated by using more careful designs of hydrophobic laminae or automated manufacturing and experimental procedures.”

REVIEWERS' COMMENTS

Reviewer #1 (Remarks to the Author):

After all the arguments in my 2 previous reports on manuscript 'Bubbles enable negative compressibility in metastable elastocapillary systems', its Authors insist that the 'modern' definition of compressibility does not require that the compressed system be thermodynamically closed and in one phase. Caprini et al. cite several definitions of 'compressibility' in the literature.

When I looked at the literature indicated by Authors in their response, I must say that I fully agree with these references! All they stress that the compressibility is defined for the assembly contained within a closed boundary, transferable only by heat and energy (and not any type of mass particles!), separating it from the thermodynamic surrounding. In these references indicated by Authors, in the same pages or in preceding pages it is stressed that the molar amounts and chemical potentials cannot change. Sometimes it is indicated directly and sometimes it is assumed as an obvious requirement and omitted. But the point is that when we calculate compressibility we cannot independently change any other thermodynamic parameter, intensive or extensive, and that the system should be in equilibrium. Composition X is one of the most fundamental thermodynamic parameters. Can we ignore it? If we could pump other materials into an assembly, then the negative volume compressibility would not be strange at all – the negative compressibility of zeolites and MOFs would have been accepted ages ago!

Compressibility remains a most fundamental characteristic of materials and this definition cannot be modified. Authors provide no evidence of negative compressibility at all, but they claim it in the title and throughout the manuscript. Therefore the refereed manuscript will only confuse these bases of thermodynamics and therefore it should not be accepted.

Reviewer #2 (Remarks to the Author):

The authors have satisfactorily addressed all points raised in my reports. The work is rigorous and innovative. I recommend publication in Nature Communications and anticipate that this work will have very high impact.

Reviewer #3 (Remarks to the Author):

The present revised version of the manuscript "Metastable elastocapillary systems with negative compressibility" (now entitled: "Bubbles enable negative compressibility in metastable elastocapillary systems") by Caprini et al. is a significant improvement over the original version of the manuscript. I believe that all my comments and those of the other reviewers have been responded properly. In particular, the definition of Negative linear compressibility used in this work is now clear. I only suggest a slight change in the title of the paper: "Bubbles enable volumetric negative compressibility in metastable elastocapillary subsystems," thus making clear the physical effect in the systems considered in this paper from the start. Therefore, I recommend publication of the manuscript in Nature once this comment has been considered.